# Coordination of tissue cell polarity by auxin transport and signaling

Carla Verna[†], Sree Janani Ravichandran, Megan G Sawchuk, Nguyen Manh Linh, Enrico Scarpella*

Department of Biological Sciences, University of Alberta, Edmonton, Canada

**Abstract** Plants coordinate the polarity of hundreds of cells during vein formation, but how they do so is unclear. The prevailing hypothesis proposes that GNOM, a regulator of membrane trafficking, positions PIN-FORMED auxin transporters to the correct side of the plasma membrane; the resulting cell-to-cell, polar transport of auxin would coordinate tissue cell polarity and induce vein formation. Contrary to predictions of the hypothesis, we find that vein formation occurs in the absence of PIN-FORMED or any other intercellular auxin-transporter; that the residual auxin-transport-independent vein-patterning activity relies on auxin signaling; and that a *GNOM*-dependent signal acts upstream of both auxin transport and signaling to coordinate tissue cell polarity and induce vein formation. Our results reveal synergism between auxin transport and signaling, and their unsuspected control by *GNOM* in the coordination of tissue cell polarity during vein patterning, one of the most informative expressions of tissue cell polarization in plants.

## Introduction

How the polarity of cells in a tissue is coordinated is a central question in biology. In animals, the coordination of this tissue cell polarity requires direct cell-cell communication and often cell movements (*Goodrich and Strutt, 2011*), both of which are precluded in plants by a wall that holds cells apart and in place; therefore, tissue cell polarity is coordinated differently in plants.

The formation of plant veins is an expression of such coordination of tissue cell polarity; this is most evident in developing leaves. Consider, for example, the formation of the midvein at the center of the cylindrical leaf primordium. Initially, the plasma-membrane (PM)-localized PIN-FORMED1 (PIN1) protein of Arabidopsis (*Gälweiler et al., 1998*), which catalyzes cellular efflux of the plant signal auxin (*Petrásek et al., 2006*), is expressed in all the inner cells of the leaf primordium (*Benková et al., 2003*; *Reinhardt et al., 2003*; *Heisler et al., 2005*; *Scarpella et al., 2006*; *Wenzel et al., 2007*; *Bayer et al., 2009*; *Verna et al., 2015*); over time, however, PIN1 expression becomes gradually restricted to the file of cells that will form the midvein. PIN1 localization at the PM of the inner cells is initially isotropic, but as PIN1 expression becomes restricted to the site of midvein formation, PIN1 localization becomes polarized: in the cells surrounding the developing midvein, PIN1 localization gradually changes from isotropic to medial, that is toward the developing midvein, to mediobasal; in the cells of the developing midvein, PIN1 becomes uniformly localized toward the base of the leaf primordium, where the midvein will connect to the pre-existing vasculature.

The correlation between coordination of tissue cell polarity, as expressed by the coordination of PIN1 polar localization between cells; polar auxin transport, as expressed by the auxin-transport-polarity-defining localization of PIN1 (*Wisniewska et al., 2006*); and vein formation does not seem to be coincidental. Auxin application to developing leaves induces the formation of broad expression domains of isotropically localized PIN1; such domains become restricted to the sites of auxin-induced vein formation, and PIN1 localization becomes polarized toward the pre-existing vasculature (*Scarpella et al., 2006*). Both the restriction of PIN1 expression domains and the polarization of

*For correspondence:
enrico.scarpella@ualberta.ca

Present address: [†]California Institute of Technology, Pasadena, United States

Competing interests: The authors declare that no competing interests exist.

**eLife digest** Plants, animals and other living things grow and develop over their lifetimes: for example, oak trees come from acorns and chickens begin their lives as eggs. To achieve these transformations, the cells in those living things must grow, divide and change their shape and other features.

Plants and animals specify the directions in which their cells will grow and develop by gathering specific proteins to one side of the cells. This makes one side different from all the other sides, which the cells use as an internal compass that points in one direction. To align their internal compasses, animal cells touch one another and often move around inside the body. Plant cells, on the other hand, are surrounded by a wall that keeps them apart and prevents them from moving around. So how do plant cells align their internal compasses?

Scientists have long thought that a protein called GNOM aligns the internal compasses of plant cells. The hypothesis proposes that GNOM gathers another protein, called PIN1, to one side of a cell. PIN1 would then pump a plant hormone known as auxin out of this first cell and, in doing so, would also drain auxin away from the cell on the opposite side. In this second cell, GNOM would then gather PIN1 to the side facing the first cell, and this process would repeat until all the cells' compasses were aligned.

To test this hypothesis, Verna et al. combined microscopy with genetic approaches to study how cells' compasses are aligned in the leaves of a plant called *Arabidopsis thaliana*. The experiments revealed that auxin needs to move from cell-to-cell to align the cells' compasses. However, contrary to the above hypothesis, this movement of auxin was not sufficient: the cells also needed to be able to detect and respond to the auxin that entered them. Along with controlling how auxin moved between the cells, GNOM also regulated how the cells responded to the auxin.

These findings reveal how plants specify which directions their cells grow and develop. In the future, this knowledge may eventually aid efforts to improve crop yields by controlling the growth and development of crop plants.

PIN1 localization are delayed by chemical inhibition of auxin transport (*Scarpella et al., 2006*; *Wenzel et al., 2007*), which induces vein pattern defects similar to, though stronger than, those of *pin1* mutants (*Mattsson et al., 1999*; *Sieburth, 1999*; *Sawchuk et al., 2013*). Therefore, available evidence suggests that auxin coordinates tissue cell polarity to induce vein formation, and that the coordinative and inductive property of auxin depends on the function of *PIN1* and possibly other *PIN* genes.

How auxin coordinates tissue cell polarity to induce vein formation is unclear, but the current hypothesis is that the GNOM (GN) guanine-nucleotide exchange factor for ADP-ribosylation-factor GTPases, which regulates vesicle formation in membrane trafficking, controls the cellular localization of PIN1 and possibly other auxin transporters; the resulting cell-to-cell, polar transport of auxin would coordinate tissue cell polarity and control developmental processes such as vein formation (reviewed in, e.g., *Berleth et al., 2000*; *Richter et al., 2010*; *Nakamura et al., 2012*; *Linh et al., 2018*). Here we tested this hypothesis by a combination of cellular imaging, molecular genetic analysis, and chemical induction and inhibition. Contrary to predictions of the hypothesis, we found that auxin-induced vein formation occurs in the absence of PIN proteins or any other intercellular auxin transporter; that the residual auxin-transport-independent vein-patterning activity relies on auxin signaling; and that a *GN*-dependent tissue-cell-polarizing signal acts upstream of both auxin transport and signaling.

## Results

### Testable predictions of the current hypothesis of coordination of tissue cell polarity and vein formation by auxin

The current hypothesis of how auxin coordinates tissue cell polarity to induce vein formation proposes that GN controls the cellular localization of PIN1 and possibly other auxin transporters; the resulting cell-to-cell, polar transport of auxin would coordinate tissue cell polarity and control

developmental processes such as vein formation (reviewed in, e.g., *Berleth et al., 2000*; *Richter et al., 2010*; *Nakamura et al., 2012*; *Linh et al., 2018*). The hypothesis makes three testable predictions:

1. The restriction of PIN1 expression domains and coordination of PIN1 polar localization that normally occur during vein formation (*Benková et al., 2003*; *Reinhardt et al., 2003*; *Heisler et al., 2005*; *Scarpella et al., 2006*; *Wenzel et al., 2007*; *Bayer et al., 2009*; *Sawchuk et al., 2013*; *Marcos and Berleth, 2014*; *Verna et al., 2015*) will occur abnormally, or will fail to occur altogether, during *gn* leaf development;
2. Were the vein pattern defects of *gn* the sole result of loss of polar auxin transport, auxin transport inhibition would lead to defects that fall within the vascular phenotype spectrum of *gn*;
3. Were the vascular defects of *gn* the result of abnormal polarity of auxin transport, they would depend on auxin transport; therefore, auxin transport inhibition should induce defects in *gn* that approximate those which it induces in *GN*.

Here we tested these predictions.

## Testing prediction 1: Restriction of PIN1 expression domains and coordination of PIN1 polar localization occur abnormally, or fail to occur altogether, during *gn* leaf development

We tested this prediction by imaging expression domains of PIN1::PIN1:YFP (PIN1:YFP fusion protein expressed by the *PIN1* promoter [*Xu et al., 2006*]) and cellular localization of expression of PIN1::PIN1:GFP (*Benková et al., 2003*) during leaf development in WT and in the new strong allele *gn-13* (*Supplementary file 1*).

Consistent with previous reports (*Benková et al., 2003*; *Reinhardt et al., 2003*; *Heisler et al., 2005*; *Scarpella et al., 2006*; *Wenzel et al., 2007*; *Bayer et al., 2009*; *Sawchuk et al., 2013*; *Marcos and Berleth, 2014*; *Verna et al., 2015*), in WT leaves PIN1::PIN1:YFP was expressed in all the cells at early stages of tissue development. Over time, epidermal expression became restricted to the basalmost cells, and inner tissue expression became restricted to developing veins (*Figure 1A–J*).

In *gn* leaves too, PIN1::PIN1:YFP was expressed in all the cells at early stages of tissue development, and over time epidermal expression became restricted to the basalmost cells; however, inner tissue expression failed to become restricted to developing veins and remained nearly ubiquitous even at very late stages of leaf development (*Figure 1K–O*).

Consistent with previous reports (*Benková et al., 2003*; *Reinhardt et al., 2003*; *Heisler et al., 2005*; *Scarpella et al., 2006*; *Wenzel et al., 2007*; *Bayer et al., 2009*; *Sawchuk et al., 2013*; *Marcos and Berleth, 2014*; *Verna et al., 2015*), in the cells of the second pair of vein loops ('second loop' hereafter) at early stages of its development in WT leaves, PIN1::PIN1:GFP expression was mainly localized to the side of the plasma membrane (PM) facing the midvein; in the inner cells flanking the developing loop, PIN1::PIN1:GFP expression was mainly localized to the side of the PM facing the developing loop; and in the inner cells further away from the developing loop, PIN1::PIN1:GFP expression was localized isotropically at the PM (*Figure 1C,P*). At later stages of second-loop development, by which time PIN1::PIN1:GFP expression had become restricted to the cells of the developing loop, PIN1::PIN1:GFP expression was localized to the side of the PM facing the midvein (*Figure 1D,T*).

At early stages of development of the tissue that in *gn* leaves corresponds to that from which the second loop forms in WT leaves, PIN1::PIN1:GFP was expressed uniformly in the outermost inner tissue, and expression was localized isotropically at the PM (*Figure 1Q,R*). PIN1::PIN1:GFP was expressed more heterogeneously in the innermost inner tissue, but expression remained localized isotropically at the PM, except in cells near the edge of higher-expression domains: in those cells, localization of PIN1::PIN1:GFP expression at the PM was weakly polar, but such weak cell polarities pointed in seemingly random directions (*Figure 1Q,S*).

At late stages of *gn* leaf development, heterogeneity of PIN1::PIN1:GFP expression had spread to the outermost inner tissue, but expression remained localized isotropically at the PM, except in cells near the edge of higher-expression domains: in those cells, localization of PIN1::PIN1:GFP expression at the PM was weakly polar, but such weak cell polarities pointed in seemingly random directions (*Figure 1U,V*). Heterogeneity of PIN1::PIN1:GFP expression in the innermost inner tissue

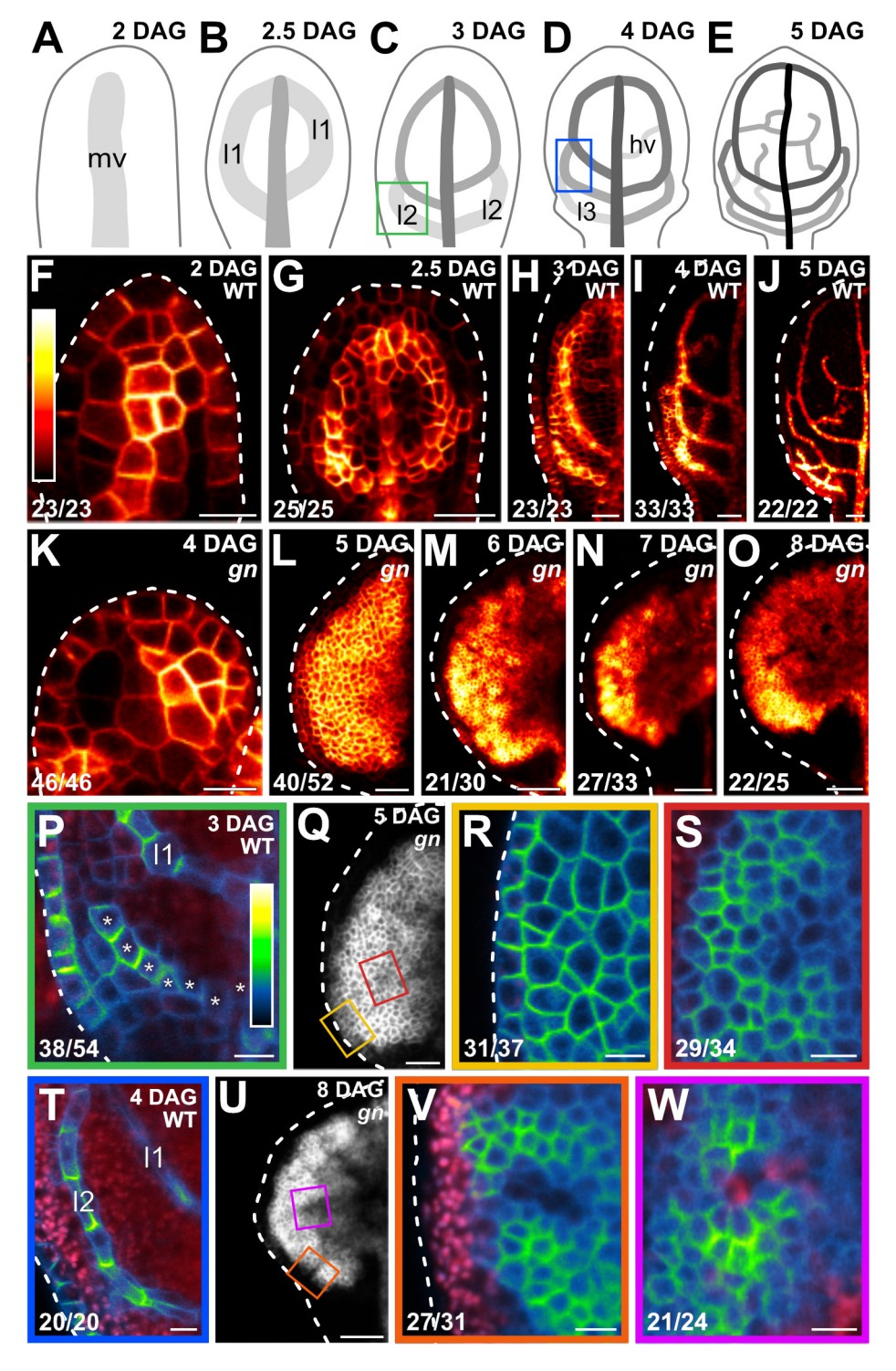

**Figure 1.** PIN1 expression and localization during *gn* leaf development. (A–Q,T,U) Top right: leaf age in days after germination (DAG). (A–E) Veins form sequentially during Arabidopsis leaf development: the formation of the midvein (mv) is followed by the formation of the first loops of veins ('first loops'; l1), which in turn is followed by the formation of second loops (l2) and minor veins (hv) (*Mattsson et al., 1999*; *Sieburth, 1999*; *Kang and Dengler, 2004*; *Scarpella et al., 2004*). Loops and minor veins differentiate in a tip-to-base sequence during leaf development. Increasingly darker grays depict progressively later stages of vein development. Boxes in C and D illustrate positions of closeups in P and T. l3: third loop. (F–W) Confocal laser scanning microscopy. First leaves.
*Figure 1 continued on next page*

*Figure 1 continued*

For simplicity, only half-leaves are shown in H–J and L–O. Dashed white line in F–R and T–V delineates leaf outline. (F–Q,T,U) Top right: genotype. (F–P,R–T,V,W) Bottom left: reproducibility index. (F–O) PIN1::PIN1:YFP expression; look-up table (ramp in F) visualizes expression levels. (P,R–T,V,W) PIN1::PIN1:GFP expression; look-up table (ramp in P) visualizes expression levels. Red: autofluorescence. Stars in P label cells of the developing second loop. (Q,U) PIN1::PIN1:YFP expression. Boxes in Q and in U illustrate positions of closeups in R and S, and in V and W, respectively. Bars: (F,P,R–T,V,W) 10 µm; (G,I,L,Q) 30 µm; (H,K) 20 µm; (J,M–O,U) 60 µm.

had become more pronounced at late stages of *gn* leaf development, and the weakly polar localization of PIN1::PIN1:GFP expression at the PM had spread to the center of the higher-expression domains (*Figure 1U,W*); nevertheless, such weak cell polarities still pointed in seemingly random directions (*Figure 1U,W*).

In conclusion, both restriction of PIN1 expression domains and coordination of PIN1 polar localization occur only to a very limited extent or fail to occur altogether during *gn* leaf development, which is consistent with the current hypothesis of how auxin coordinates tissue cell polarity to induce vein formation.

## Testing prediction 2: Auxin transport inhibition leads to defects that fall within the vascular phenotype spectrum of *gn*

### Vascular phenotype spectrum of *gn*

WT Arabidopsis grown under normal conditions forms separate leaves whose vein networks are defined by at least four reproducible features (*Telfer and Poethig, 1994*; *Nelson and Dengler, 1997*; *Kinsman and Pyke, 1998*; *Candela et al., 1999*; *Mattsson et al., 1999*; *Sieburth, 1999*; *Steynen and Schultz, 2003*; *Sawchuk et al., 2013*; *Verna et al., 2015*) (*Figure 2A,B*):

1. a narrow I-shaped midvein that runs the length of the leaf;
2. lateral veins that branch from the midvein and join distal veins to form closed loops;
3. minor veins that branch from midvein and loops, and either end freely or join other veins;
4. minor veins and loops that curve near the leaf margin, lending a scalloped outline to the vein network.

In the leaves of the new weak allele *gn-18* (*Supplementary file 1*) (*Figure 2—figure supplement 1*) closed loops were often replaced by open loops, that is loops that contact the midvein or other loops at only one of their two ends; and veins were often replaced by 'vein fragments', that is stretches of vascular elements that fail to contact other stretches of vascular elements at either one of their two ends (*Figure 2C–E,N*; *Figure 2—figure supplement 2*). Loops were open and veins were fragmented also in the leaves of both *gn^fwr* (*Okumura et al., 2013*) and *gn^B/E* (*Geldner et al., 2004*) (*Figure 2N*; *Figure 2—figure supplement 2*). In addition, the vein network of *gn^B/E* leaves was denser (*Figure 2F,N*; *Figure 2—figure supplement 2*).

The vein network was denser also in all the leaves of the intermediate alleles *gn^R5* (*Geldner et al., 2004*), *gn^van7* (*Koizumi et al., 2000*) and *gn^van7+fwr;gn-13*, in which we had combined the *van7* and *fwr* mutations (*Supplementary file 1*) (*Figure 2G,N*; *Figure 2—figure supplement 2*). However, in the leaves of these backgrounds, unlike in those of *gn^B/E*, the veins were thicker; lateral veins failed to join the midvein but ran parallel to it to form a 'wide midvein'; and the vein network outline was jagged because of narrow clusters of vascular elements that were oriented perpendicular to the leaf margin and that were laterally connected by veins (*Figure 2G,H,K,N*; *Figure 2—figure supplement 2*).

In most of the leaves of the strong alleles *gn^SALK_103014* (*Okumura et al., 2013*), *gn-13* and *gn^emb30-8* (*Franzmann et al., 1989*; *Moriwaki et al., 2014*), a central, shapeless vascular cluster was connected with the basal part of the leaf by a wide midvein; vascular elements were oriented seemingly randomly at the distal side of the cluster and gradually more parallel to the leaf axis toward the proximal side of the cluster (*Figure 2I,L–N*; *Figure 2—figure supplement 2*). In the remaining leaves of these backgrounds, vascular differentiation was limited to a central, shapeless cluster of seemingly randomly oriented vascular elements (*Figure 2J,M,N*; *Figure 2—figure supplement 2*).

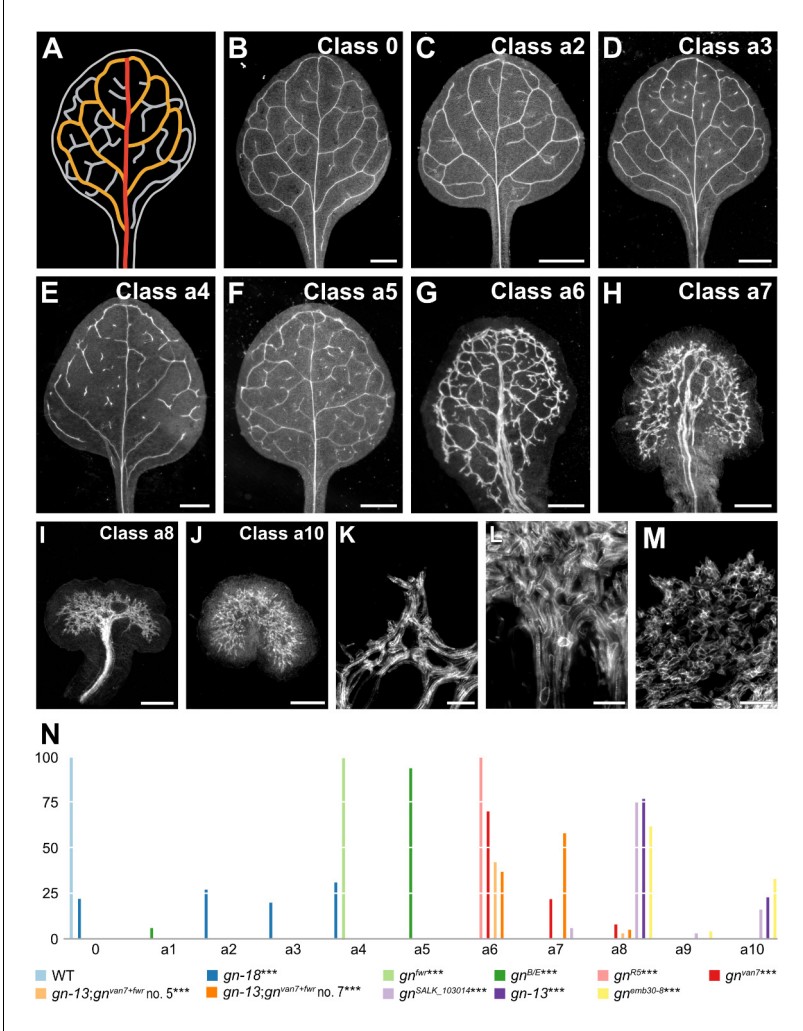

**Figure 2.** Vascular phenotype spectrum of *gn*. (A,B) Vein pattern of WT mature first leaf. In A: red, midvein; orange, loops; gray, minor veins. (B–J) Dark-field illumination of mature first leaves illustrating phenotype classes (top right): class 0, narrow I-shaped midvein and scalloped vein-network outline (B); class a1, dense vein network and apically thickened vein-network outline (not shown); class a2, open vein-network outline (C); class a3, fragmented vein network (D); class a4, open vein-network outline and fragmented vein network (E); class a5, open vein-network outline, fragmented vein network, and apically thickened vein-network outline (F); class a6, wide midvein, dense network of thick veins, and jagged vein-network outline (G); class a7, dense network of thick veins that fail to join the midvein in the bottom half of the leaf and pronouncedly jagged vein-network outline (H); class a8, wide midvein and shapeless vascular cluster (I); class a9, fused leaves with wide midvein and shapeless vascular cluster (not shown); class a10, shapeless vascular cluster (J). (K–M) Details of vascular clusters illustrating vascular elements uniformly oriented perpendicular to the leaf margin (K) (class a6); vascular elements oriented seemingly randomly at the distal side of the cluster and parallel to the leaf axis at the proximal side of the cluster (L) (classes a8 and a9); and seemingly random orientation of vascular elements (M) (classes a8–a10). (N) Percentages of leaves in phenotype classes. Difference between *gn-18* and WT, between $gn^{fwr}$ and WT, between $gn^{B/E}$ and WT, between $gn^{R5}$ and WT, between $gn^{van7}$ and WT, between $gn^{van7+fwr}$;*gn-13* and WT, between $gn^{SALK\_103014}$ and WT, between *gn-13* and WT, and between *emb30-8* and WT was significant at $p < 0.001$ (\*\*\*) by Kruskal-Wallis and Mann-Whitney test with Bonferroni correction. Sample population sizes: WT, 66; *gn-18*, 172; $gn^{fwr}$, 43; $gn^{B/E}$, 85; $gn^{R5}$, 93; $gn^{van7}$, 109; $gn^{van7+fwr}$;*gn-13* no. 5, 97; $gn^{van7+fwr}$;*gn-13* no. 7, 93; $gn^{SALK\_103014}$, 32; *gn-13*, 56; $gn^{emb30-8}$, 45. Bars: (B–F) 1 mm; (G) 0.75 mm; (H,I) 0.5 mm; (J) 0.25 mm; (K–M) 50 µm. See *Figure 2—figure supplement 1* for effect of the *gn-18* mutation on *GN* expression. See *Figure 2—figure supplement 2* for alternative visual display of distribution of leaves in phenotype classes.

The online version of this article includes the following source data and figure supplement(s) for figure 2:

**Source data 1.** Distribution and frequency of leaves in phenotype classes and statistical analysis.

*Figure 2 continued*

**Figure supplement 1.** Effect of the *gn-18* mutation on *GN* expression.
**Figure supplement 2.** Percentages of leaves in phenotype classes.

## Vein pattern defects induced by auxin transport inhibition

### Vein pattern defects of *pin* mutants

Five of the eight PIN proteins of Arabidopsis (*Paponov et al., 2005*; *Krecek et al., 2009*; *Viaene et al., 2013*) — PIN1, PIN2, PIN3, PIN4, and PIN7 (hereafter collectively referred to as PM-PIN) — are primarily localized to the PM and catalyze cellular auxin efflux (*Chen et al., 1998*; *Gälweiler et al., 1998*; *Luschnig et al., 1998*; *Müller et al., 1998*; *Friml et al., 2002a*; *Friml et al., 2002b*; *Friml et al., 2003*; *Petrásek et al., 2006*; *Yang and Murphy, 2009*; *Zourelidou et al., 2014*). *pin1* is the only *pin* single mutant with vein pattern defects, and the vein pattern defects of double mutants between *pin1* and *pin2*, *pin3*, *pin4*, or *pin7* are no different from those of *pin1* single mutants (*Sawchuk et al., 2013*), suggesting that either *PIN2*, *PIN3*, *PIN4*, and *PIN7* have no function in *PIN1*-dependent vein patterning or their function in this process is redundant. To discriminate between these possibilities, we first assessed the collective contribution to *PIN1*-dependent vein patterning of *PIN3*, *PIN4*, and *PIN7*, whose translational fusions to GFP (*Zadnikova et al., 2010*; *Bennett et al., 2016*; *Belteton et al., 2018*) (*Supplementary file 1*) are all expressed, as are translational fusions of PIN1 to GFP (*Benková et al., 2003*; *Heisler et al., 2005*; *Scarpella et al., 2006*; *Wenzel et al., 2007*; *Bayer et al., 2009*; *Marcos and Berleth, 2014*), in both epidermal and inner cells of the developing leaf (*Figure 3A,C–E*).

Consistent with previous reports (*Sawchuk et al., 2013*; *Verna et al., 2015*), the vein patterns of most of the *pin1* leaves were abnormal (*Figure 3F,G,L*; *Figure 3—figure supplement 1A*). *pin3; pin4;pin7* (*pin3;4;7* hereafter) embryos were viable and developed into seedlings (*Supplementary file 2A*) (*Figure 3—figure supplement 2A*) whose vein patterns were no different from those of WT (*Figure 3L*; *Figure 3—figure supplement 1A*). *pin1,3;4;7* embryos were also viable (*Supplementary file 2B*) and developed into seedlings (*Supplementary file 2C*) (*Figure 3—figure supplement 2A,B*; *Figure 3—figure supplement 3A–H*) whose vein pattern defects were more severe than those of *pin1* (*Figure 3H–L*; *Figure 3—figure supplement 1A*); however, as in WT, in *pin1,3;4;7* vascular elements were still aligned along the length of the vein (*Figure 3J,K*).

We next asked whether mutation of *PIN2*, whose translational fusion to GFP (*Xu and Scheres, 2005*) is only expressed in epidermal cells in the developing leaf (*Figure 3B*), changed the spectrum of vein pattern defects of *pin1,3;4;7*.

*pin2;3;4;7* embryos were viable and developed into seedlings (*Supplementary file 2A*) (*Figure 3—figure supplement 3A*) whose vein patterns were no different from those of WT (*Figure 3L*; *Figure 3—figure supplement 1A*). *pin1,3;2;4;7* embryos were also viable (*Supplementary file 2B*) and developed into seedlings (*Supplementary file 2C*) (*Figure 3—figure supplement 2A–C*; *Figure 3—figure supplement 3A–H*) whose vein pattern defects were no different from those of *pin1,3;4;7* (*Figure 3L*; *Figure 3—figure supplement 1A*).

The three remaining PIN proteins of Arabidopsis — PIN5, PIN6, and PIN8 — are primarily localized to the endoplasmic reticulum (ER) (*Mravec et al., 2009*; *Bosco et al., 2012*; *Ding et al., 2012*; *Sawchuk et al., 2013*). *PIN6* and *PIN8*, but not *PIN5*, provide vein patterning functions that overlap with those of *PIN1* (*Sawchuk et al., 2013*; *Verna et al., 2015*). We asked what the collective contribution to vein patterning were of the auxin transport pathway defined by PIN6 and PIN8, and of that defined by PIN1, PIN3, PIN4, and PIN7.

As previously reported (*Sawchuk et al., 2013*), the vein pattern of *pin6;8* was no different from that of WT (*Figure 3O*; *Figure 3—figure supplement 1B*). *pin1,3,6;4;7;8* embryos were viable (*Supplementary file 2B*) and developed into seedlings (*Supplementary file 2C*) (*Figure 3—figure supplement 2A,B,D*; *Figure 3—figure supplement 3A–H*) whose vein patterns differed from those of *pin1,3;4;7* in four respects (*Figure 3H,I,L,M–O*; *Figure 3—figure supplement 1B*):

1. The vein network comprised more lateral veins;
2. Lateral veins failed to join the midvein but ran parallel to it to form a wide midvein;
3. Lateral veins ended in a marginal vein that closely paralleled the leaf margin, lending a smooth outline to the vein network;

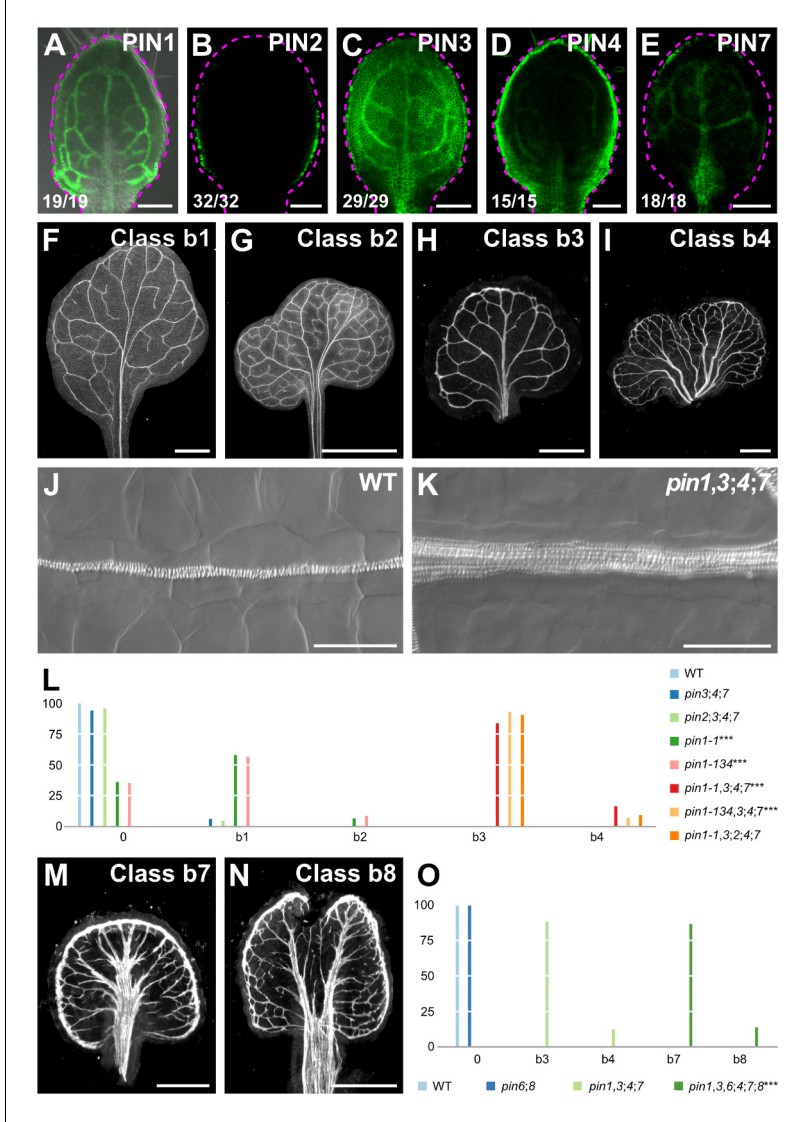

**Figure 3.** Vein pattern defects of *pin* mutants. (A–K,M,N) Top right: expression-reported gene, phenotype class, or genotype. (B–E) Bottom left: reproducibility index. (A–E) Confocal laser scanning microscopy with (A) or without (B–E) transmitted light; 4-day-old first leaves. Dashed magenta line delineates leaf outline. (A) PIN1::PIN1:GFP expression. (B) PIN2::PIN2:GFP expression. (C) PIN3::PIN3:GFP expression. (D) PIN4::PIN4:GFP expression. (E) PIN7::PIN7:GFP expression. (F–I,M,N) Dark-field illumination images of mature first leaves illustrating phenotype classes: class b1, Y-shaped midvein and scalloped vein-network outline (F); class b2, fused leaves with scalloped vein-network outline (G); class b3, thick veins and scalloped vein-network outline (H); class b4, fused leaves with thick veins and scalloped vein-network outline (I); class b7, wide midvein, more lateral-veins, and conspicuous marginal vein (M); class b8, fused leaves with wide midvein, more lateral-veins, and conspicuous marginal vein (N). (J,K) Differential interference images of details of WT (J) or *pin1-1,3;4;7* (K) illustrating normal (classes 0, b1, and b2) or thick (classes b3 and b4) veins, respectively. (L,O) Percentages of leaves in phenotype classes (Class 0 defined in *Figure 2*). (L) Difference between *pin1-1* and WT, between *pin1-134* and WT, between *pin1-1,3;4;7* and *pin1-1*, and between *pin1-134,3;4;7* and *pin1-134* was significant at p<0.001 (\*\*\*) by Kruskal-Wallis and Mann-Whitney test with Bonferroni correction. Sample population sizes: WT, 58; *pin2;3;4;7*, 49; *pin3;4;7*, 102; *pin1-1*, 81; *pin1-134*, 48; *pin1-1,3;4;7*, 75; *pin1-134,3;4;7*, 45; *pin1-1,3;2;4;7*, 99. (O) Difference between *pin1-1,3,6;4;7;8* and *pin1-1,3;4;7* was significant at p<0.001 (\*\*\*) by Kruskal-Wallis and Mann-Whitney test with Bonferroni correction. Sample population sizes: WT, 51; *pin6;8*, 47; *pin1-1,3;4;7*, 49; *pin1-1,3,6;4;7;8*, 73. Bars: (A–E) 0.1 mm; (F–H) 1 mm; (I) 5 mm; (J,K) 50 µm; (M,N) 0.5 mm. See *Figure 3—figure supplement 1* for alternative visual display of distribution of leaves in phenotype classes. See *Figure 3—figure supplement 2* for *pin* mutant seedlings. See *Figure 3—figure supplement 3* for cotyledon patterns of *pin* mutants.

*Figure 3 continued on next page*

*Figure 3 continued*

The online version of this article includes the following source data and figure supplement(s) for figure 3:

**Source data 1.** Distribution and frequency in phenotype classes and statistical analysis of the leaves in *Figure 3L* and *Figure 3—figure supplement 1A*.

**Source data 2.** Distribution and frequency in phenotype classes and statistical Analysis of the Leaves in *Figure 3O* and *Figure 3—figure supplement 1B*.

**Figure supplement 1.** Percentages of leaves in phenotype classes.

**Figure supplement 2.** *pin* mutant seedlings.

**Figure supplement 3.** Cotyledon patterns of *pin* mutants.

4. Veins were thicker.

In conclusion, *PIN3*, *PIN4*, and *PIN7* provide no nonredundant function in vein patterning but collectively contribute to *PIN1*-dependent vein patterning; *PIN2* seems to have no function in this process; and the auxin-transport pathway mediated by PIN1, PIN3, PIN4, and PIN7, and that mediated by PIN6 and PIN8 provide overlapping functions in vein patterning. Most important, loss of *PIN*-dependent vein-patterning function fails to lead to defects that fall within the vascular phenotype spectrum of *gn*.

## Vein pattern defects induced by chemical inhibition of auxin transport

Cellular auxin efflux is inhibited by a class of structurally related compounds exemplified by N-1-naphthylphthalamic acid (NPA) (*Cande and Ray, 1976*; *Katekar and Geissler, 1980*; *Sussman and Goldsmith, 1981*). Because PM-PIN proteins catalyze cellular auxin efflux (*Chen et al., 1998*; *Petrásek et al., 2006*; *Yang and Murphy, 2009*; *Zourelidou et al., 2014*), we asked whether defects resulting from simultaneous mutation of all the *PM-PIN* genes with vein patterning function were phenocopied by growth of WT in the presence of NPA. To address this question, we compared defects of *pin1,3;4;7* with those induced in WT by growth in the presence of NPA.

The vein patterns of *pin1,3;4;7* lacked all the characteristic defects induced in WT by NPA (*Figure 4A,B,D,E,H*; *Figure 4—figure supplement 1*). However, such defects were induced in *pin1,3;4;7* by NPA (*Figure 4F,H*; *Figure 4—figure supplement 1*), suggesting that this background has residual NPA-sensitive vein-patterning activity. The vein pattern defects induced in WT or *pin1,3;4;7* by NPA were no different from those of *pin1,3,6;4;7;8* (*Figure 4C,D–F,H*; *Figure 4—figure supplement 1*). Because no additional defects were induced in *pin1,3,6;4;7;8* by NPA (*Figure 4G,H*; *Figure 4—figure supplement 1*), the residual NPA-sensitive vein-patterning activity of *pin1,3;4;7* is provided by *PIN6* and *PIN8*.

In conclusion, growth in the presence of NPA phenocopies defects of loss of *PIN*-dependent vein patterning function; in the absence of this function, any residual NPA-sensitive vein-patterning activity — if existing — becomes inconsequential; and neither loss of *PIN*-dependent vein-patterning function nor loss of NPA-sensitive vein-patterning activity leads to defects that fall within the vascular phenotype spectrum of *gn*.

## Vascular phenotype of *abcb* mutants

Cellular auxin efflux is catalyzed not only by PM-PIN proteins but by the PM-localized ATP-BINDING CASSETTE B1 (ABCB1) and ABCB19 proteins (*Geisler et al., 2005*; *Bouchard et al., 2006*; *Petrásek et al., 2006*; *Blakeslee et al., 2007*; *Yang and Murphy, 2009*), whose fusions to GFP (*Dhonukshe et al., 2008*; *Mravec et al., 2008*) are expressed at early stages of leaf development (*Figure 5A,B*). We asked whether ABCB1/19-mediated auxin efflux were required for vein patterning.

The embryos of *abcb1* and *abcb19* were viable, but ~15% of *abcb1;19* embryos died during embryogenesis (*Supplementary file 2D*); nevertheless, the vein patterns of *abcb1*, *abcb19*, and *abcb1;19* were no different from the vein pattern of WT (*Figure 5E,F,I*; *Figure 5—figure supplement 1A*), suggesting that ABCB1/19-mediated auxin efflux is dispensable for vein patterning.

Functions of ABCB1/19-mediated auxin transport overlap with those of PIN-mediated auxin transport (*Blakeslee et al., 2007*; *Mravec et al., 2008*). We therefore asked whether vein pattern defects

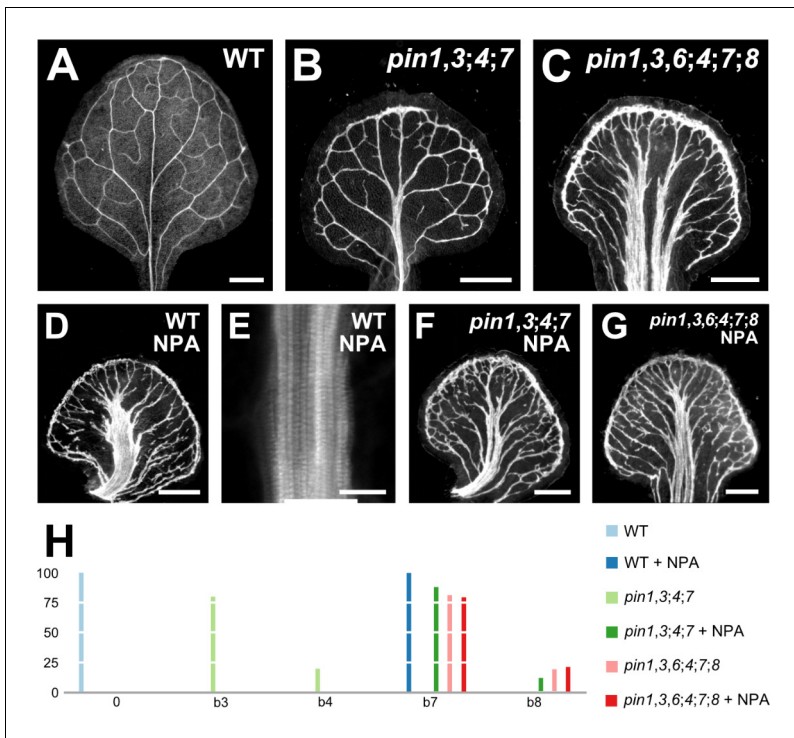

**Figure 4.** Vein pattern defects induced by chemical inhibition of auxin transport. (**A–G**) Top right: genotype and treatment. Dark-field illumination (**A–D,F,G**) or confocal laser scanning microscopy (**E**) of mature first leaves. (**E**) Detail illustrating thick veins in NPA-grown WT (compare with *Figure 3J*). (**H**) Percentages of leaves in phenotype classes (defined in *Figures 2* and *3*). Sample population sizes: WT, 38; *pin1-1,3;4;7*, 30; *pin1-1,3,6;4;7;8*, 73; NPA-grown WT, 41; NPA-grown *pin1-1,3;4;7*, 58; NPA-grown *pin1-1,3,6;4;7;8*, 48. Bars: (**A–D,F,G**) 0.5 mm, (**E**) 25 µm. See *Figure 4—figure supplement 1* for alternative visual display of distribution of leaves in phenotype classes.

The online version of this article includes the following source data and figure supplement(s) for figure 4:

**Source data 1.** Distribution and frequency of leaves in phenotype classes and statistical analysis.
**Figure supplement 1.** Percentages of leaves in phenotype classes.

resulting from simultaneous mutation of *PIN1*, *PIN3*, and *PIN6*, or induced in WT by NPA were enhanced by simultaneous mutation of *ABCB1* and *ABCB19*.

*pin1,3,6* embryos were viable (*Supplementary file 2E*) and developed into seedlings (*Supplementary file 2F*) (*Figure 5—figure supplement 2*). The proportion of embryos derived from the self-fertilization of *PIN1,PIN3,PIN6/pin1,pin3,pin6;abcb1/abcb1;abcb19/abcb19* that died during embryogenesis was no different from the proportion of embryos derived from the self-fertilization of *abcb1/abcb1;abcb19/abcb19* that died during embryogenesis (*Supplementary file 2E*), suggesting no nonredundant functions of *PIN1*, *PIN3*, and *PIN6* in *ABCB1/ABCB19*-dependent embryo viability. The vein pattern defects of *pin1,3,6;abcb1;19* were no different from those of *pin1,3,6*, and the vein pattern defects induced in *abcb1;19* by NPA were no different from those induced in WT by NPA (*Figure 5C,D,G–I*; *Figure 5—figure supplement 1A*), suggesting no vein-patterning function of *ABCB1* and *ABCB19* in the absence of function of *PIN1*, *PIN3*, and *PIN6*, or of NPA-sensitive, *PIN*-dependent vein-patterning function.

Vein patterning functions of ABCB1/19-mediated auxin efflux might be masked by redundant functions provided by other ABCB transporters. The TWISTED DWARF1/ULTRACURVATA2 (TWD1/UCU2; TWD1 hereafter) protein (*Kamphausen et al., 2002*; *Pérez-Pérez et al., 2004*) is a positive regulator of ABCB-mediated auxin transport (*Geisler et al., 2003*; *Bouchard et al., 2006*; *Bailly et al., 2008*; *Wu et al., 2010*; *Wang et al., 2013*). Consistent with this observation, defects of *twd1* are more severe than, though similar to, those of *abcb1;19* (*Geisler et al., 2003*;

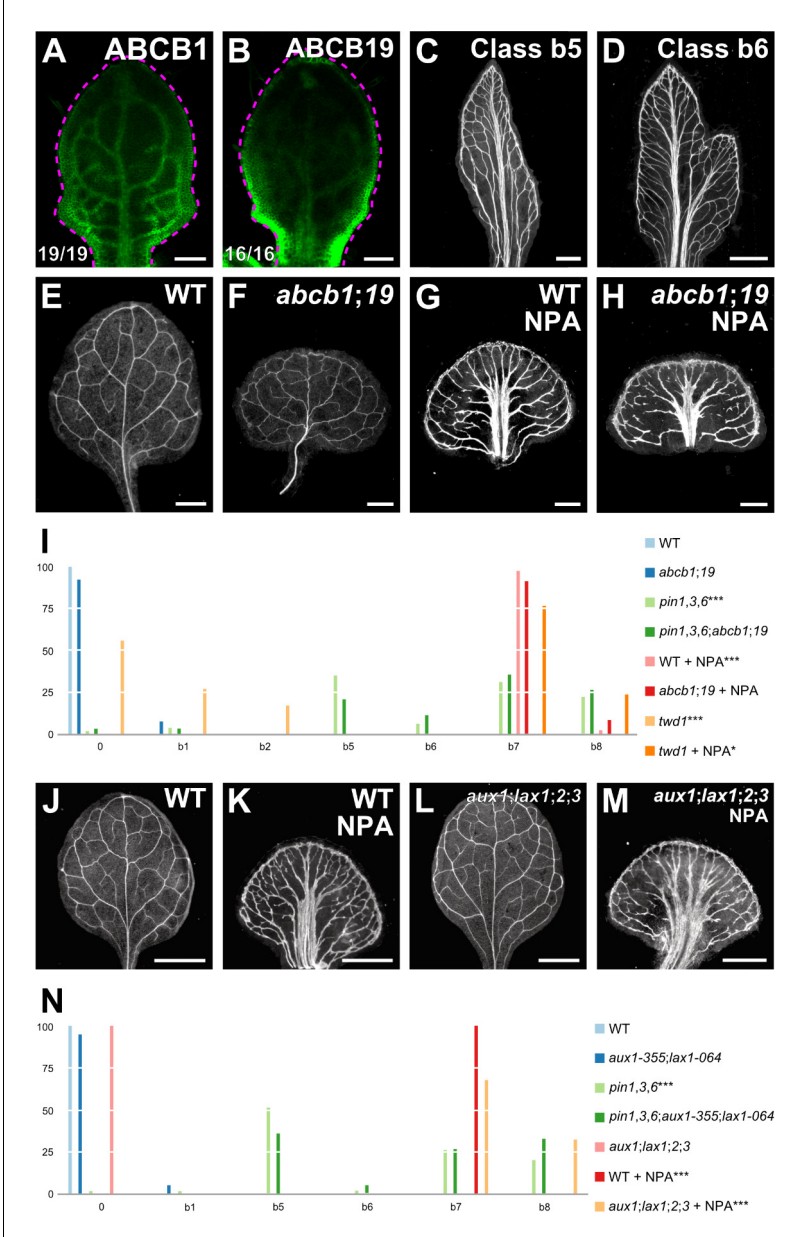

**Figure 5.** Vascular phenotype of *abcb* and *aux1/lax* mutants. (A,B,E–H,J–M) Top right: expression-reported gene, genotype, and treatment. (A,B) Bottom left: reproducibility index. (A,B) Confocal laser scanning microscopy; 5-day-old first leaves. Dashed magenta line delineates leaf outline. (A) ABCB1::ABCB1:GFP expression. (B) ABCB19::ABCB19:GFP expression. (C–H,J–M) Dark-field illumination of mature first leaves. (C,D) Phenotype classes: class b5, thick veins and conspicuous marginal vein (C); class b6, fused leaves with thick veins and conspicuous marginal vein (D). (I,N) Percentages of leaves in phenotype classes (Classes 0, b1, b2, b7, and b8 defined in *Figures 2* and *3*). Difference between *pin1-1,3,6* and WT, between *twd1* and WT, and between NPA-grown WT and WT was significant at p<0.001 (***); and between NPA-grown *twd1* and NPA-grown WT was significant at p<0.05 (*) by Kruskal-Wallis and Mann-Whitney test with Bonferroni correction. Sample population sizes: WT, 41; *abcb1;19*, 40; *pin1-1,3,6*, 80; *pin1-1,3,6;abcb1;19*, 62; NPA-grown WT, 43; NPA-grown *abcb1;19*, 46; *twd1*, 41; NPA-grown *twd1*, 46. (N) Difference between *pin1-1,3,6* and WT, between NPA-grown WT and WT, and between NPA-grown *aux1-21;lax1;2;3* and NPA-grown WT was significant at p<0.001 (***) by Kruskal-Wallis and Mann-Whitney test with Bonferroni correction. Sample population sizes: WT, 53; *aux1-21;lax1;2;3*, 60; *aux1-355; lax1-064*, 77; *pin1-1,3,6*, 75; *pin1-1,3,6;aux1-355;lax1-064*, 58; NPA-grown WT, 46; NPA-grown *aux1-21;lax1;2;3*, 40. Bars: (A,B) 0.1 mm; (C–H) 0.5 mm.; (J–M) 1 mm. See *Figure 5—figure supplement 1* for alternative visual display of distribution of leaves in phenotype classes. See *Figure 5—figure supplement 2* for cotyledon patterns of *pin*,

*Figure 5 continued on next page*

*Figure 5 continued*

*abcb*, and *aux1/lax* mutants. See *Figure 5—figure supplement 3* for effect of the *aux1-355* mutation on *AUX1* expression and of the *lax1-064* mutation on *LAX1* expression.

The online version of this article includes the following source data and figure supplement(s) for figure 5:

**Source data 1.** Distribution and frequency in phenotype classes and statistical analysis of the leaves in *Figure 5I* and *Figure 5—figure supplement 1A*.
**Source data 2.** Distribution and frequency in phenotype classes and statistical analysis of the leaves in *Figure 5N* and *Figure 5—figure supplement 1B*.
**Figure supplement 1.** Percentages of leaves in phenotype classes.
**Figure supplement 2.** Cotyledon patterns of *pin*, *abcb,* and *aux1/lax* mutants.
**Figure supplement 3.** Effect of the *aux1-355* Mutation on *AUX1* Expression and of the *lax1-064* Mutation on *LAX1* Expression.

*Bouchard et al., 2006*; *Bailly et al., 2008*; *Wu et al., 2010*; *Wang et al., 2013*). We therefore reasoned that analysis of *twd1* vein patterns may uncover vein patterning functions of ABCB-mediated auxin transport that could not be inferred from the analysis of *abcb1;19*.

Though some of the *twd1* leaves had vein pattern defects (*Figure 5I*; *Figure 5—figure supplement 1A*), the vein pattern defects induced in *twd1* by NPA were no different from those induced in WT or *abcb1;19* by NPA (*Figure 5I*; *Figure 5—figure supplement 1A*), suggesting that vein patterning functions of *TWD1*-dependent ABCB-mediated auxin transport become inconsequential in the absence of NPA-sensitive, *PIN*-dependent vein-patterning function.

Therefore, the residual vein patterning activity in *pin* mutants or in their NPA-induced phenocopy is not provided by *ABCB1*, *ABCB19* or *TWD1*-dependent ABCB-mediated auxin transport, and loss of PIN- and ABCB-mediated auxin transport fails to lead to defects that fall within the vascular phenotype spectrum of *gn*.

## Vascular phenotype of *aux1/lax* mutants

Auxin is predicted to enter the cell by diffusion and through an auxin influx carrier (*Rubery and Sheldrake, 1974*; *Raven, 1975*). In Arabidopsis, auxin influx activity is encoded by the *AUX1*, *LAX1*, *LAX2*, and *LAX3* (*AUX1/LAX*) genes (*Parry et al., 2001*; *Yang et al., 2006*; *Swarup et al., 2008*; *Péret et al., 2012*). We asked whether AUX1/LAX-mediated auxin transport were required for vein patterning.

*aux1;lax1;2;3* embryos were viable (*Supplementary file 2G*). Because the vein patterns of *aux1; lax1;2;3* were no different from those of WT (*Figure 5J,L,N*; *Figure 5—figure supplement 1B*), we conclude that *AUX1/LAX* function is dispensable for vein patterning.

We next asked whether contribution of *AUX1/LAX* genes to vein patterning only became apparent in conditions of extremely reduced PIN-mediated auxin transport. To address this question, we tested whether vein pattern defects resulting from simultaneous loss of function of *PIN1*, *PIN3*, and *PIN6*, or induced in WT by NPA were enhanced by simultaneous mutation of *AUX1* and *LAX1* — the two *AUX1/LAX* genes that most contribute to shoot organ patterning (*Bainbridge et al., 2008*) (*Supplementary file 1*) (*Figure 5—figure supplement 3*) — or of all *AUX1/LAX* genes, respectively.

The embryos derived from the self-fertilization of *PIN1,pin3,PIN6/pin1,pin3,pin6;aux1/aux1;lax1/ lax1* were viable (*Supplementary file 2H*) and developed into seedlings (*Supplementary file 2I*) (*Figure 5—figure supplement 2*). The vein pattern defects of *pin1,3,6;aux1;lax1* were no different from those of *pin1,3,6* (*Figure 5N*; *Figure 5—figure supplement 1B*), and the vein pattern defects induced in *aux1;lax1;2;3* by NPA were no different from those induced in WT by NPA (*Figure 5K,M, N*; *Figure 5—figure supplement 1B*), suggesting no vein-patterning function of *AUX1/LAX* genes in conditions of extremely reduced auxin transport.

Therefore, the residual vein patterning activity in *pin* mutants or in their NPA-induced phenocopy is not provided by *AUX1/LAX* genes, and loss of PIN- and AUX1/LAX-mediated auxin transport fails to lead to defects that fall within the vascular phenotype spectrum of *gn*.

## Comparing the vein pattern defects induced by auxin transport inhibition with the vascular phenotype spectrum of *gn*

Auxin transport inhibition leads to defects that are qualitatively different from and quantitatively weaker than those of *gn* (*Figures 2– 4*). Therefore, our results fail to support Prediction 2 of the current hypothesis of how auxin coordinates tissue cell polarity to induce vein formation. Consequently, the hypothesis must be revised.

### Testing prediction 3: Auxin transport inhibition induces defects in *gn* that approximate those which it induces in *GN*

To test this prediction, we first asked what the phenotype were of the quintuple mutant between the strong allele *gn-13* (*Figure 2*) and mutation in *PIN1*, *PIN3*, *PIN4*, and *PIN7* — that is the *PM-PIN* genes with vein patterning function (*Figure 3*).

*gn;pin1,3;4;7* embryos were viable (*Supplementary file 2J*) and developed into seedlings (*Supplementary file 2K*) whose cotyledon and leaf vascular defects were no different from those of *gn* (*Figure 6A,B,E*; *Figure 6—figure supplement 1*; *Figure 3—figure supplement 2A,B*; *Figure 3—figure supplement 3*; *Figure 6—figure supplement 2A-D*; *Figure 6—figure supplement 3A-F*; *Figure 6—figure supplement 4*; *Figure 6—figure supplement 5A-D,F-H,K*), suggesting that the vascular phenotype of *gn* is epistatic to that of *pin1,3;4;7*.

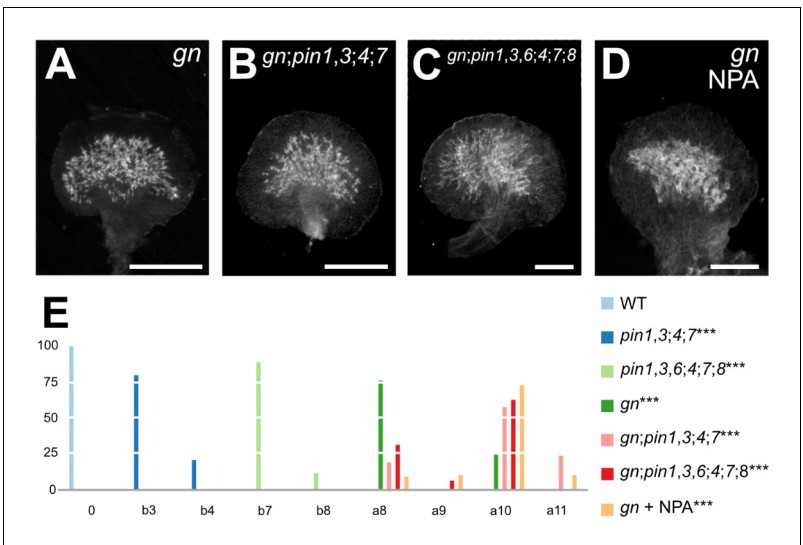

**Figure 6.** Vascular Defects of auxin-transport-inhibited *gn*. (A–D) Dark-field illumination of mature first leaves. Top right: genotype and treatment. (E) Percentages of leaves in phenotype classes (Classes 0, a8–a10, b3, b4, b7, and b8 defined in *Figures 2* and *3*; class a11, fused leaves with shapeless vascular cluster — not shown). Difference between *pin1-1,3;4;7* and WT, between *pin1-1,3,6;4;7;8* and WT, between *gn* and WT, between *gn-13;pin1-1,3;4;7* and *pin1-1,3;4;7*, between *gn-13;pin1-1,3,6;4;7;8* and *pin1-1,3,6;4;7;8*, and between NPA-grown *gn-13* and *pin1-1,3,6;4;7;8* was significant at p<0.001 (***) by Kruskal-Wallis and Mann-Whitney test with Bonferroni correction. Sample population sizes: WT, 63; *pin1-1,3;4;7*, 53; *pin1-1,3,6;4;7;8*, 52; *gn-13*, 69; *gn-13;pin1-1,3;4;7*, 21; *gn-13;pin1-1,3,6;4;7;8*, 16; NPA-grown *gn-13*, 60. Bars: (A–D) 0.5 mm. See *Figure 6—figure supplement 1* for alternative visual display of distribution of leaves in phenotype classes. See *Figure 6—figure supplement 2* for *pin* and *gn* mutant seedlings. See *Figure 6—figure supplement 3* for seedling axes of *pin* and *gn* mutants. See *Figure 6—figure supplement 4* for cotyledon patterns of *pin* and *gn* mutants. See *Figure 6—figure supplement 5* for cotyledon vein patterns of *pin* and *gn* mutants.

The online version of this article includes the following source data and figure supplement(s) for figure 6:

**Source data 1.** Distribution and frequency of leaves in phenotype classes and statistical analysis.
**Figure supplement 1.** Percentages of leaves in phenotype classes.
**Figure supplement 2.** *pin* and *gn* mutant seedlings.
**Figure supplement 3.** Seedling axes of *pin* and *gn* mutants.
**Figure supplement 4.** Cotyledon patterns of *pin* and *gn* mutants.
**Figure supplement 5.** Cotyledon vein patterns of *pin* and *gn* mutants.

We next asked what the phenotype were of the septuple mutant between the strong allele *gn-13* (*Figure 2*) and mutation in all the *PIN* genes with vein patterning function (*Figure 3*).

*gn;pin1,3,6;4;7;8* embryos were viable (*Supplementary file 2J*) and developed into seedlings (*Supplementary file 2K*) whose cotyledon and leaf vascular defects were no different from those of *gn* (*Figure 6A,C,E*; *Figure 6—figure supplement 1*; *Figure 3—figure supplement 2A,D*; *Figure 3— figure supplement 3*; *Figure 6—figure supplement 2A,C,E,F*; *Figure 6—figure supplement 3A,B, F-I*; *Figure 6—figure supplement 4*; *Figure 6—figure supplement 5A-G,I-K*), suggesting that the vascular phenotype of *gn* is epistatic to that of *pin1,3,6;4;7;8*. Finally, NPA failed to induce additional vein pattern defects in *gn* leaves (*Figure 6D,E*; *Figure 6—figure supplement 1*).

In conclusion, auxin transport inhibition fails to induce defects in *gn* that approximate those which it induces in *GN*. Therefore, our results also fail to support Prediction 3 of the current hypothesis of how auxin coordinates tissue cell polarity to induce vein formation. Consequently, the hypothesis must be revised.

## Revising the current hypothesis of coordination of tissue cell polarity and vein formation by auxin

### Auxin-induced vein formation in the absence of auxin transport

The uniform vein-pattern phenotype of *pin1,3,6;4;7;8* was phenocopied by growth of WT in the presence of NPA (*Figure 4*). Moreover, the vein pattern phenotype of *pin1,3,6;4;7;8* was unchanged by NPA treatment, and the NPA-induced vein-pattern phenocopy of *pin1,3,6;4;7;8* was unchanged by mutation in any other intercellular auxin-transporter (*Figures 4* and *5*). These observations suggest that the vein pattern phenotype of *pin1,3,6;4;7;8* or of its NPA-induced phenocopy is symptomatic of absence of that component of auxin transport that is relevant to vein patterning (see also Discussion). Because auxin transport is thought to be essential for auxin-induced vascular-strand formation (reviewed in *Sachs, 1981*; *Berleth et al., 2000*; *Aloni, 2010*; *Sawchuk and Scarpella, 2013*), we asked whether auxin induced vein formation in *pin1,3,6;4;7;8* and, consequently, whether veins were formed by an auxin-dependent mechanism in *pin1,3,6;4;7;8*. To address this question, we applied the natural auxin indole-3-acetic acid (IAA) to one side of developing leaves of WT and *pin1,3,6;4;7;8*, and recorded tissue response in mature leaves.

Consistent with previous reports (*Scarpella et al., 2006*; *Sawchuk et al., 2007*), in most of the WT leaves IAA induced formation of extra veins (*Figure 7A,B*). IAA induced the formation of extra veins in *pin1,3,6;4;7;8* leaves too (*Figure 7C,D*), but it also induced the formation of tissue outgrowths of varied shape; nevertheless, IAA induced vascular strand formation in most of those tissue outgrowths (*Figure 7—figure supplement 1*).

We conclude that *pin1,3,6;4;7;8* leaves respond to vein-formation-inducing auxin signals and, consequently, that veins are formed by an auxin-dependent mechanism in the absence of that component of auxin transport that is relevant to vein patterning.

### Auxin-signaling-dependent vein patterning in the absence of auxin transport

Leaves of *pin1,3,6;4;7;8* respond to vein-formation-inducing auxin signals (*Figure 7*), suggesting that the residual vein-patterning activity in those leaves may be provided by an auxin-dependent mechanism. We therefore asked what the contribution of auxin signaling to vein patterning were in the absence of *PIN*-dependent vein patterning activity — that is of that component of auxin transport that is relevant to vein patterning. To address this question, we used mutants in *AUXIN-RESISTANT1* (*AXR1*), which lack a required post-translational modification of the auxin receptor complex (reviewed in *Calderon-Villalobos et al., 2010*; *Schwechheimer, 2018*); double mutants in *TRANS-PORT INHIBITOR RESPONSE1* (*TIR1*) and *AUXIN SIGNALING F-BOX2* (*AFB2*), which lack the two auxin receptors that most contribute to auxin signaling (*Dharmasiri et al., 2005*); and phenylboronic acid (PBA), which inhibits auxin signaling (*Matthes and Torres-Ruiz, 2016*).

The embryos of *axr1* and *tir1;afb2* were viable (*Supplementary file 2L*) and developed into seedlings whose vein pattern defects were similar to those of weak *gn* alleles (*Figure 2*) — loops were open and veins were fragmented. Similar defects were observed in WT grown in the presence of PBA (*Figure 8A,B,H*; *Figure 8—figure supplement 1*).

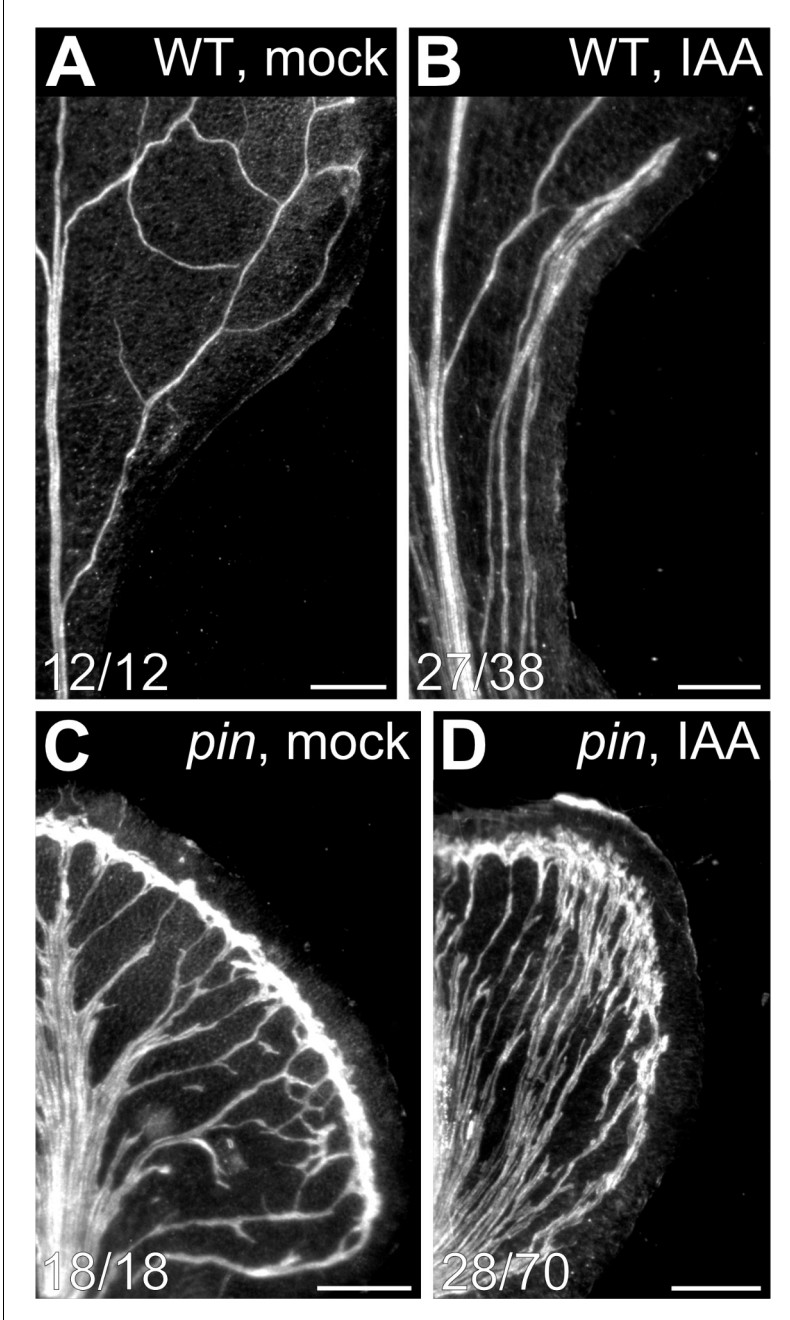

**Figure 7.** Auxin-induced vein formation in *pin* mutants. (A–D) Top right: genotype and treatment. Dark-field illumination of mature first leaves of WT (A,B) or *pin1-1,3,6;4;7;8* (C,D) at side of application of lanolin paste (A,C) or lanolin paste containing 1% IAA (B,D). Bars: (A) 0.5 mm; (B–D) 0.25 mm. See *Figure 7—figure supplement 1* for additional effects of auxin application to *pin* mutants.

The online version of this article includes the following figure supplement(s) for figure 7:

**Figure supplement 1.** Auxin-induced formation of vascularized tissue outgrowths in *pin* mutants.

We next asked whether PBA, *axr1*, or *tir1;afb2* enhanced the vein pattern defects induced by NPA or by mutation in all the *PIN* genes with vein patterning function.

A few of the leaves of NPA-grown *axr1*, NPA-grown *tir1;afb2*, and NPA- and PBA-grown WT resembled those of NPA-grown WT or of *pin1,3,6;4;7;8* (*Figure 4*; *Figure 8C,H*; *Figure 8—figure supplement 1*). However, many of the leaves of NPA-grown *axr1*, NPA-grown *tir1;afb2*, and NPA-

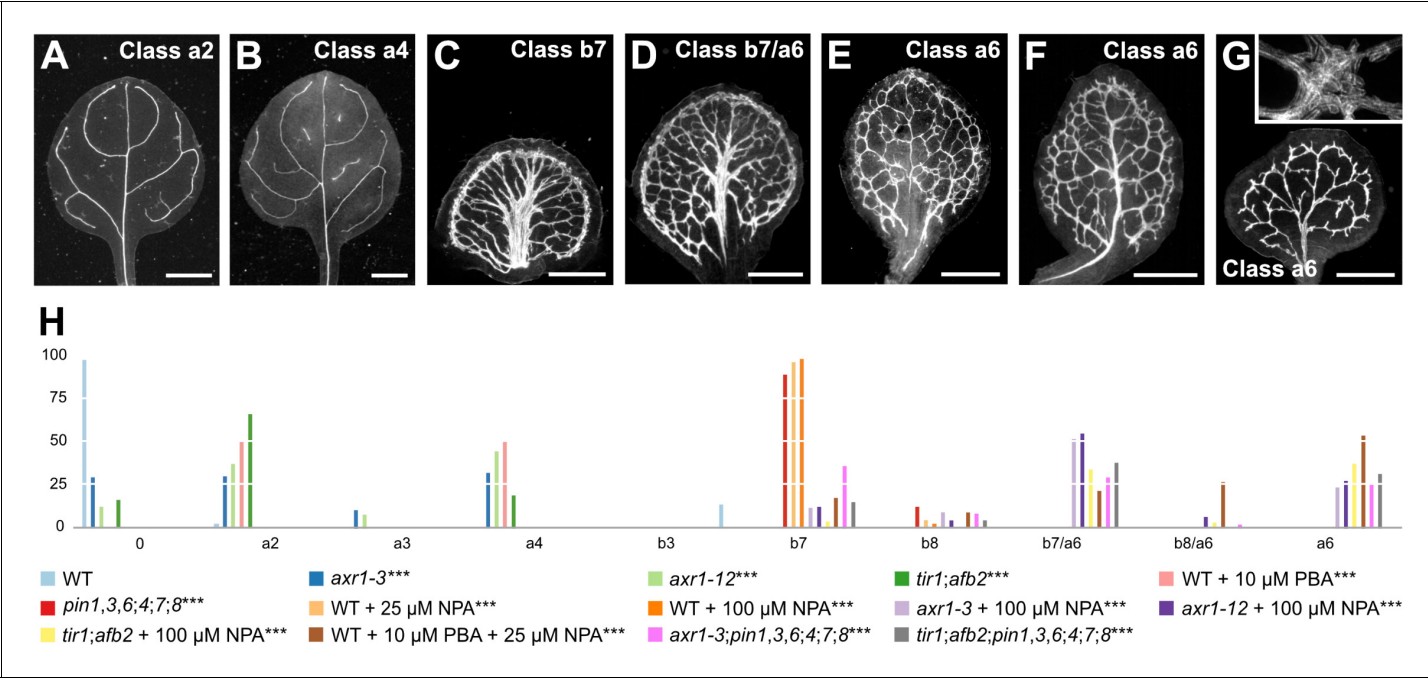

**Figure 8.** Auxin-signaling-dependent vein patterning in the absence of auxin transport. (A–G) Dark-field illumination of mature leaves illustrating phenotype classes (A–F, top right; G, bottom left): class a2 (*axr1-3*; A); class a4 (*tir1;afb2*; B); class b7 (NPA-grown WT; C); class b7/a6, wide midvein, more lateral-veins, dense network of thick veins, and conspicuous marginal vein (NPA-grown *axr1-12*; D); class b8/a6, fused leaves with wide midvein, more lateral-veins, dense network of thick veins, and conspicuous marginal vein (not shown); class a6 (E: PBA- and NPA-grown WT; F: NPA-grown *tir1; afb2*; G: *tir1;afb2;pin1-1,3,6;4;7;8*); inset in (G) illustrates cluster of seemingly randomly oriented vascular elements. (H) Percentages of leaves in phenotype classes (Classes 0, a2–a4, a6, b3, b7, and b8 defined in *Figures 2* and *3*). Difference between *axr1-3* and WT, between *axr1-12* and WT, between *tir1;afb2* and WT, between PBA-grown WT and WT, between *pin1-1,3,6;4;7;8* and WT, between NPA-grown WT and WT, between NPA-grown *axr1-3* and NPA-grown WT, between NPA-grown *axr1-12* and NPA-grown WT, between NPA-grown *tir1;afb2* and NPA-grown WT, between PBA- and NPA-grown WT and NPA-grown WT, between *axr1-3;pin1-1,3,6;4;7;8* and *pin1-1,3,6;4;7;8*, and between *tir1;afb2;pin1-1,3,6;4;7;8* and *pin1-1,3,6;4;7;8* was significant at p<0.001 (***) by Kruskal-Wallis and Mann-Whitney test with Bonferroni correction. Sample population sizes: WT, 47; *axr1-3*, 41; *axr1-12*, 41; *tir1;afb2*, 42; PBA-grown WT, 55; *pin1-1,3,6;4;7;8*, 43; NPA-grown WT, 48 (25 μM) or 146 (100 μM); NPA-grown *axr1-3*, 101; NPA-grown *axr1-12*, 103; NPA-grown *tir1;afb2*, 65; PBA- and NPA-grown WT, 105; *axr1-3;pin1-1,3,6;4;7;8*, 62; *tir1;afb2;pin1-1,3,6;4;7;8*, 75. Bars: (A,B) 1 mm; (C–E) 0.75 mm (F,G) 0.5 mm. See *Figure 8—figure supplement 1* for alternative visual display of distribution of leaves in phenotype classes. See *Figure 8—figure supplement 2* for *pin* and *axr1* mutant seedlings. See *Figure 8—figure supplement 3* for cotyledon patterns of *pin*, *axr1*, and *tir1;afb2* mutants. See *Figure 8—figure supplement 4* for *pin* and *tir1;afb2* mutant seedlings.

The online version of this article includes the following source data and figure supplement(s) for figure 8:

**Source data 1.** Distribution and frequency of leaves in phenotype classes and statistical analysis.

**Figure supplement 1.** Percentages of leaves in phenotype classes.

**Figure supplement 2.** *pin* and *axr1* mutant seedlings.

**Figure supplement 3.** Cotyledon patterns of *pin*, *axr1*, and *tir1;afb2* mutants.

**Figure supplement 4.** *pin* and *tir1;afb2* mutant seedlings.

and PBA-grown WT resembled those of intermediate *gn* alleles (*Figure 2*): veins were thicker; the vein network was denser; and its outline was jagged because of narrow clusters of vascular elements that were oriented perpendicular to the leaf margin and that were laterally connected by veins or that, in the most severe case, were aligned in seemingly random orientations (*Figure 8E–H*; *Figure 8—figure supplement 1*). The remaining leaves of NPA-grown *axr1*, NPA-grown *tir1;afb2*, and NPA- and PBA-grown WT had features intermediate between those of NPA-grown WT or of *pin1,3,6;4;7;8* and those of intermediate *gn* alleles (*Figures 2* and *4*; *Figure 8D,H*; *Figure 8—figure supplement 1*). Finally, the embryos of *axr1;pin1,3,6;4;7;8* and *tir1;afb2;pin1-1,3,6;4;7;8* were viable (*Supplementary file 2M*) and developed into seedlings (*Supplementary file 2N*) (*Figure 8—figure supplements 2–4*) whose vein pattern defects were no different from those of NPA-grown *axr1* and NPA-grown *tir1;afb2* (*Figure 8C–H*; *Figure 8—figure supplement 1*).

These observations suggest that the residual vein-patterning activity in *pin1,3,6;4;7;8* is provided, at least in part, by AXR1- and TIR1/AFB2-mediated auxin signaling. Because reduction of AXR1- and TIR1/AFB2-mediated auxin signaling enhanced vein pattern defects resulting from loss of *PIN*-dependent vein-patterning function, we conclude that PIN-mediated auxin transport and AXR1- and TIR1/AFB2-mediated auxin signaling provide overlapping functions in vein patterning. Finally, the similarity between the vein pattern defects of NPA-grown *axr1* and *tir1;afb2*, of NPA- and PBA-grown WT, and of *axr1;pin1,3,6;4;7;8* and *tir1;afb2;pin1,3,6;4;7;8*, on the one hand, and those of intermediate *gn* alleles, on the other, suggests that the vein pattern defects of *gn* are caused by simultaneous defects in auxin transport and signaling.

## Control of auxin-signaling-dependent vein patterning by *GN*

Were the vascular defects of *gn* not only the result of abnormal polarity or loss of PIN-mediated auxin transport but that of defects in auxin signaling, the vein pattern defects of *gn* would be associated with reduced auxin response, and the reduced auxin response of *gn* would be recapitulated by NPA-grown *axr1*. To test whether that were so, we imaged expression of the auxin response reporter DR5rev::nYFP (*Heisler et al., 2005*; *Sawchuk et al., 2013*) in developing leaves of WT, *pin1,3,6;4;7;8*, NPA-grown WT, *axr1*, *gn*, and NPA-grown *axr1*.

As previously shown (*Sawchuk et al., 2013*; *Verna et al., 2015*), strong DR5rev::nYFP expression was mainly associated with developing veins in WT (*Figure 9A*). In *pin1,3,6;4;7;8* and NPA-grown WT, DR5rev::nYFP expression was weaker and mainly confined to areas near the margin of the leaf (*Figure 9B,C*; *Figure 9—figure supplement 1*). DR5rev::nYFP expression was weaker also in *axr1* but was still associated with developing veins (*Figure 9D*; *Figure 9—figure supplement 1*). Finally, in both *gn* and NPA-grown *axr1*, DR5rev::nYFP expression was much weaker and scattered across large areas of the leaf (*Figure 9E,F*; *Figure 9—figure supplement 1*), suggesting that the vein pattern defects of *gn* are associated with reduced auxin response and that the reduced auxin response of *gn* is recapitulated by NPA-grown *axr1*.

Were the vascular defects of *gn* caused by simultaneous defects in auxin transport and signaling, and did *GN* control auxin signaling as it controls auxin transport, the vascular defects of *gn;axr1* would resemble those of *gn*, just as the vascular defects of *gn;pin1,3;4;7* and *gn;pin1,3,6;4;7;8* resemble those of *gn*; we tested whether that were so.

*gn;axr1* embryos were viable (*Supplementary file 2O*) and developed into seedlings (*Supplementary file 2P*) (*Figure 9—figure supplements 2* and *3*) whose vascular defects were no different from those of *gn* (*Figure 9G–I*; *Figure 9—figure supplement 4*), suggesting that the phenotype of *gn* is epistatic to that of *axr1*.

We conclude that the vascular defects of *gn* are caused by simultaneous defects in auxin transport and signaling, and that *GN* controls both auxin signaling and auxin transport.

## Coordination of tissue cell polarity by *GN*-dependent auxin transport and signaling

The vein pattern defects of *gn* are caused by simultaneous defects in auxin transport and signaling (*Figures 8* and *9*). We finally asked whether simultaneous defects in auxin transport and signaling recapitulated *gn* defects in coordination of tissue cell polarity. To address this question, we imaged cellular localization of PIN1::PIN1:GFP expression during leaf development in WT, *tir1;afb2*, NPA-grown WT, *gn*$^{van7}$, and NPA-grown *tir1;afb2*.

Consistent with previous reports (*Benková et al., 2003*; *Reinhardt et al., 2003*; *Heisler et al., 2005*; *Scarpella et al., 2006*; *Wenzel et al., 2007*; *Bayer et al., 2009*; *Sawchuk et al., 2013*; *Marcos and Berleth, 2014*; *Verna et al., 2015*), and as shown above (*Figure 1P,T*), in the cells of the second loop at early stages of its development in WT leaves, PIN1::PIN1:GFP expression was mainly localized to the side of the PM facing the midvein; in the inner cells flanking the developing loop, PIN1::PIN1:GFP expression was mainly localized to the side of the PM facing the developing loop; and in the inner cells further away from the developing loop, PIN1::PIN1:GFP expression was localized isotropically at the PM (*Figure 10B*). At later stages of second-loop development, by which time PIN1::PIN1:GFP expression had become restricted to the cells of the developing loop, PIN1::PIN1:GFP expression was localized to the side of the PM facing the midvein (*Figure 10H*). We observed a similar pattern of localization of PIN1::PIN1:GFP expression in *tir1;afb2*, but in this

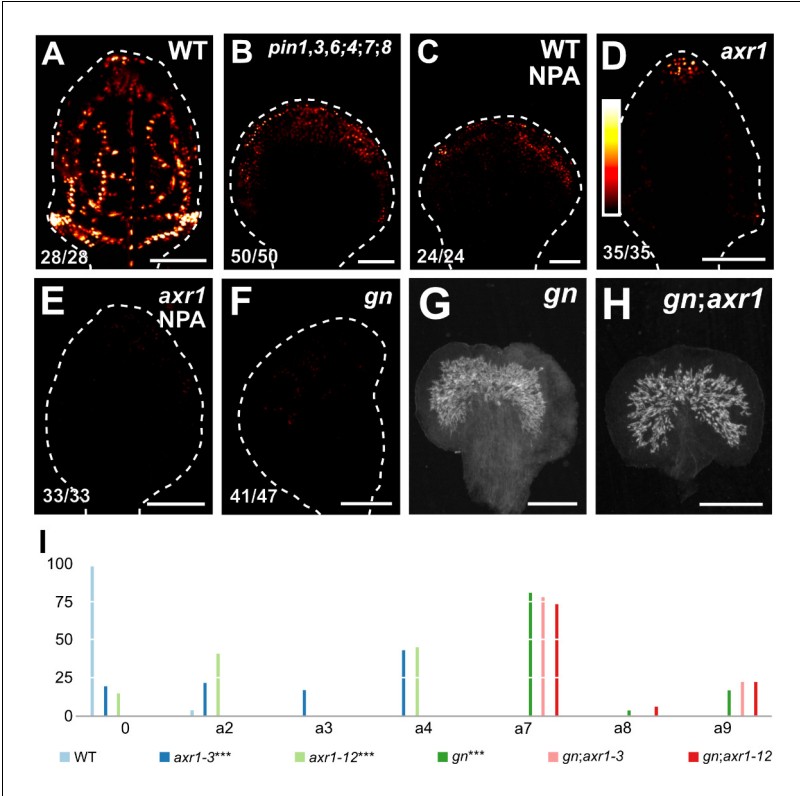

**Figure 9.** Auxin-signaling-dependent vascular development in *gn*. (A–F) Confocal laser scanning microscopy; first leaves 4 (A,C), 5 (B,D,E) or 6 (F) days after germination. DR5rev::nYFP expression; look-up table (ramp in D) visualizes expression levels. Top right: genotype and treatment. Bottom left: reproducibility index. Dashed white line delineates leaf outline. Images were taken at identical settings. (G,H) Dark-field illumination of mature first leaves. Top right: genotype. (I) Percentages of leaves in phenotype classes (defined in *Figure 2*). Difference between *axr1-3* and WT, between *axr1-12* and WT, and between *gn-13* and WT was significant at p<0.001 (***) by Kruskal-Wallis and Mann-Whitney test with Bonferroni correction. Sample population sizes: WT, 49; *axr1-3*, 42; *axr1-12*, 49; *gn-13*, 47; *gn-13;axr1-3*, 45; *gn-13;axr1-12*, 45. Bars: (A–F) 100 μm; (G,H) 0.75 mm. See *Figure 9—figure supplement 1* for images of DR5rev::nYFP expression taken by matching signal intensity to detector's input range (~5% saturated pixels). See *Figure 9—figure supplement 2* for *gn* and *axr1* mutant seedlings. See *Figure 9—figure supplement 3* for cotyledon patterns of *gn* and *axr1* mutants. See *Figure 9—figure supplement 4* for alternative visual display of distribution of leaves in phenotype classes.

The online version of this article includes the following source data and figure supplement(s) for figure 9:

**Source data 1.** Distribution and frequency of leaves in phenotype classes and statistical analysis.
**Figure supplement 1.** Auxin response in developing leaves.
**Figure supplement 2.** *gn* and *axr1* mutant seedlings.
**Figure supplement 3.** Cotyledon patterns of *gn* and *axr1* mutants.
**Figure supplement 4.** Percentages of leaves in phenotype classes.

background stages of second-loop development comparable to those in WT appeared at later stages of leaf development, and most of the second loops failed to connect to the first loop (*Figure 10C,I*).

Consistent with previous reports (*Scarpella et al., 2006*; *Wenzel et al., 2007*), PIN1::PIN1:GFP expression domains were broader at early stages of development of the tissue that in NPA-grown WT corresponds to that from which the second loop forms in WT; PIN1::PIN1:GFP expression was localized isotropically at the PM in the outermost inner cells but was mainly localized to the basal side of the PM in the innermost inner cells (*Figure 10D*). At later stages of second-loop development in NPA-grown WT, by which time PIN1::PIN1:GFP expression had become restricted to the cells of the developing loop, PIN1::PIN1:GFP expression was localized to the basal side of the PM (*Figure 10J*).

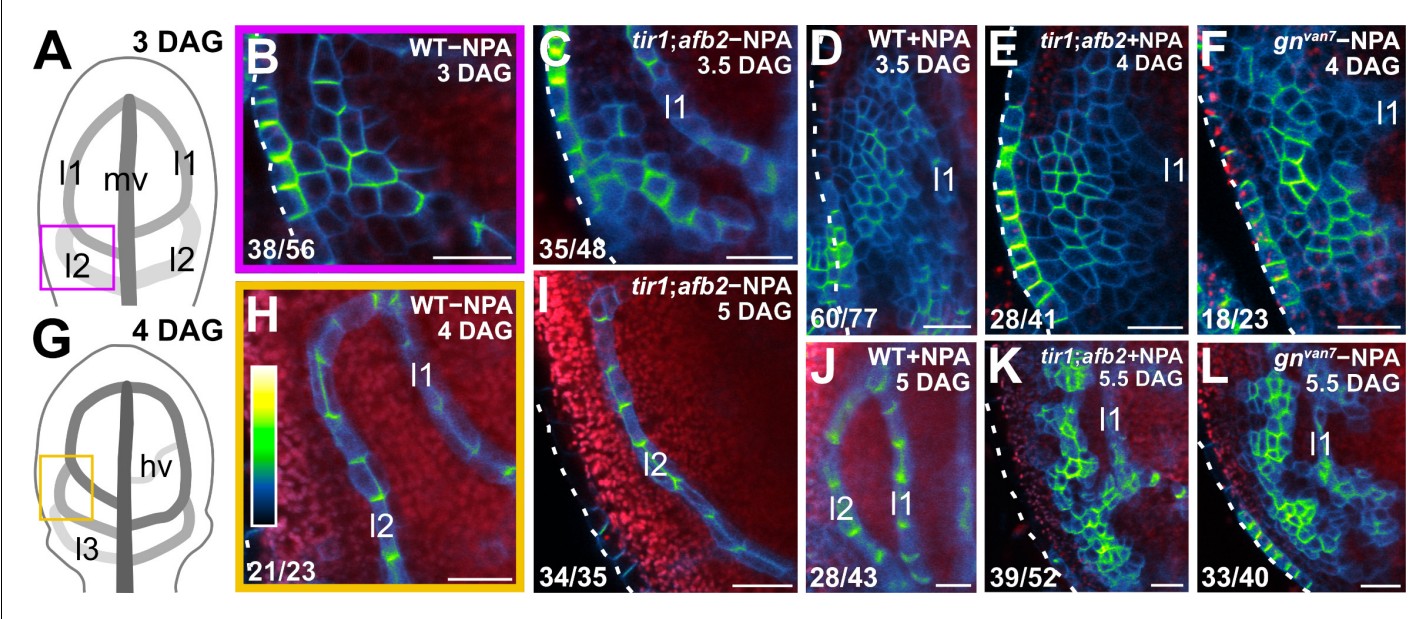

**Figure 10.** Auxin-transport- and auxin-signaling-dependent coordination of PIN1 localization in *gn* developing leaves. (A,G) Increasingly darker grays depict progressively later stages of vein development. Boxes illustrate positions of closeups in B and H, respectively. hv: minor vein; l1, l2 and l3: first, second and third loops; mv: midvein. (B–F,H–L) Confocal laser scanning microscopy. First leaves. Top right: genotype, treatment and leaf age in days after germination (DAG). Dashed white line delineates leaf outline. Bottom left: reproducibility index. PIN1::PIN1:GFP expression; look-up table (ramp in H) visualizes expression levels. Red: autofluorescence. (I) 24/35 of second loops failed to connect to the first loop. Bars: (B–F,H–L) 20 µm.

As in NPA-grown WT, in both *gn^van7* and NPA-grown *tir1;afb2* PIN1::PIN1:GFP expression domains were broader at early stages of development of the tissue that corresponds to that from which the second loop forms in WT, but PIN1::PIN1:GFP was expressed more heterogeneously in *gn^van7* and NPA-grown *tir1;afb2* than in NPA-grown WT (*Figure 10E,F*). Nevertheless, as in NPA-grown WT, in both *gn^van7* and NPA-grown *tir1;afb2* PIN1::PIN1:GFP expression remained localized isotropically at the PM, except in cells near the edge of higher-expression domains: in those cells, localization of PIN1::PIN1:GFP expression at the PM was weakly polar, but such weak cell polarities pointed in seemingly random directions (*Figure 10E,F*).

At later stages of second-loop development of both *gn^van7* and NPA-grown *tir1;afb2*, heterogeneity of PIN1::PIN1:GFP expression had become more pronounced, and PIN1::PIN1:GFP expression had become restricted to narrow clusters of cells; in those cells, localization of PIN1::PIN1:GFP expression at the PM was weakly polar, but such weak cell polarities still pointed in seemingly random directions (*Figure 10K,L*).

In conclusion, simultaneous defects in auxin transport and signaling recapitulate *gn* defects in coordination of PIN1 polar localization, suggesting not only that the vein pattern defects of *gn* are caused by simultaneous defects in auxin transport and signaling, but that simultaneous defects in auxin transport and signaling recapitulate *gn* defects in coordination of tissue cell polarity during vein formation.

## Discussion

The current hypothesis of how auxin coordinates tissue cell polarity to induce vein formation proposes that GN controls the cellular localization of PIN1 and other PIN proteins; the resulting cell-to-cell, polar transport of auxin would coordinate tissue cell polarity and control developmental processes such as vein formation (reviewed in, e.g., *Berleth et al., 2000*; *Richter et al., 2010*; *Nakamura et al., 2012*; *Linh et al., 2018*).

Contrary to predictions of the hypothesis, we find that auxin-induced vein formation occurs in the absence of PIN proteins or any other intercellular auxin transporter; that the residual auxin-transport-independent vein-patterning activity relies on auxin signaling; and that a *GN*-dependent signal

that coordinates tissue cell polarity to induce vein formation acts upstream of both auxin transport and signaling (*Figure 11*).

## Control of vein patterning by polar auxin transport

Overwhelming experimental evidence suggests that the patterned formation of veins depends on polar auxin transport (reviewed in *Sachs, 1981*; *Sachs, 1991*; *Berleth et al., 2000*; *Sachs, 2000*; *Sawchuk and Scarpella, 2013*). The polarity of auxin transport is determined by the asymmetric localization of efflux carriers of the PIN family at the PM of auxin-transporting cells (*Wisniewska et al., 2006*). Therefore, loss of function of all the PM-PIN proteins should lead to loss of reproducible vein-pattern features or even, in the most extreme case, to the inability to form veins. Neither prediction is, however, supported by evidence: mutants in all the *PM-PIN* genes with vein patterning function — *PIN1*, *PIN3*, *PIN4* and *PIN7* — or in all the *PM-PIN* genes — *PIN1–PIN4* and *PIN7* — form veins, and these veins are arranged in reproducible, albeit abnormal, patterns. We conclude that vein patterning is controlled by additional, PM-PIN-independent auxin-transport pathways.

The existence of PM-PIN-independent auxin-transport pathways with vein patterning function can also be inferred from the discrepancy between the vein pattern defects of *pin1,3;4;7* or *pin1,3;2;4;7* and those induced by NPA, which is thought to be a specific inhibitor of cellular auxin efflux (*Cande and Ray, 1976*; *Sussman and Goldsmith, 1981*; *Petrásek et al., 2003*; *Dhonukshe et al., 2008*). The vein pattern defects of WT grown in the presence of NPA are more severe than those of *pin1,3;4;7* or *pin1,3;2;4;7*, suggesting the existence of an NPA-sensitive auxin-transport pathway with vein patterning function besides that controlled by PM-PIN proteins, a suggestion that is supported by the ability of NPA to enhance the vein pattern defects of *pin1,3;4;7* to match those induced in WT by NPA.

Such PM-PIN-independent NPA-sensitive auxin-transport pathway with vein patterning function depends on the activity of the ER-PIN proteins PIN6 and PIN8, as inferred from the identity of the vein pattern defects induced in WT by NPA and those of *pin1,3,6;4;7;8*, and from the inability of NPA to induce further defects in *pin1,3,6;4;7;8*. Moreover, that NPA-grown WT phenocopies *pin1,3,6;4;7;8*; that no further defects can be induced in *pin1,3,6;4;7;8* by NPA; and that the vein

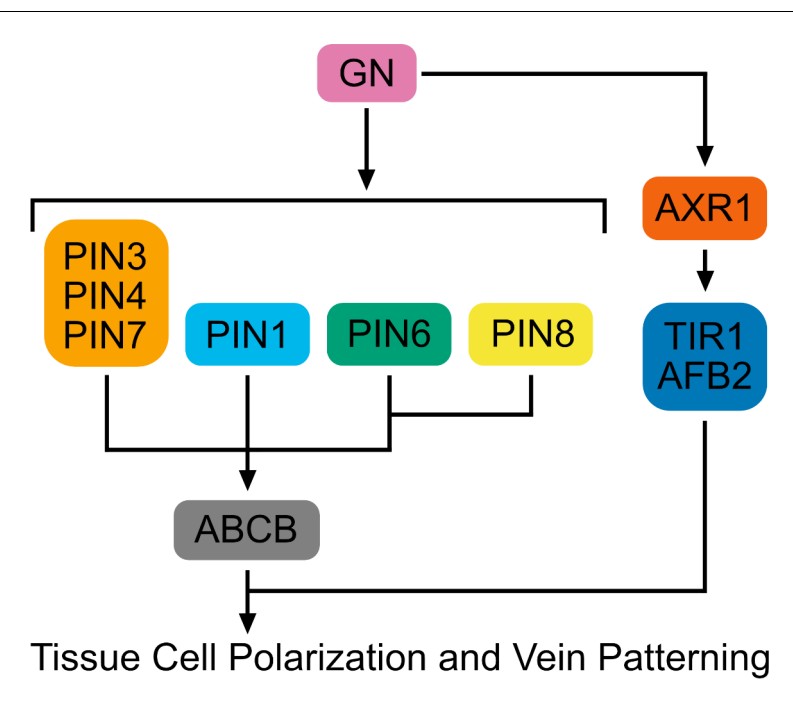

**Figure 11.** Interpretation summary. Genetic interaction network controlling tissue cell polarization and vein patterning. Arrows indicate positive effects.

patterns of *pin1,3,6;4;7;8* and NPA-grown WT fall into the same single phenotype-class suggest no NPA-sensitive vein-patterning activity beyond that provided by PIN1, PIN3, PIN4, PIN6, PIN7, and PIN8, and hence the existence of NPA-insensitive vein-patterning pathways.

These NPA-insensitive vein-patterning pathways unlikely depend on the function of other intercellular auxin transporters — the AUX1/LAX influx carriers (*Yang et al., 2006*; *Swarup et al., 2008*; *Péret et al., 2012*) and the ABCB efflux carriers (*Geisler et al., 2005*; *Bouchard et al., 2006*; *Petrásek et al., 2006*) — as their mutation fails to enhance the vein pattern defects of *pin1,3,6* and of the NPA-induced phenocopy of *pin1,3,6;4;7;8*. The NPA-insensitive vein-patterning pathways also unlikely depend on NPA-insensitive auxin transport because as little as 10 µM NPA (a fraction of the concentration we used) is sufficient to inhibit polar auxin transport completely in tissue segments (*Okada et al., 1991*; *Kaneda et al., 2011*). Whatever the molecular nature of the NPA-insensitive vein-patterning pathways, they do contribute to the polar propagation of the inductive auxin signal: application of auxin to *pin1,3,6;4;7;8* leaves, just as to WT leaves, induces the formation of veins that connect the applied auxin to the pre-existing vasculature basal to the site of auxin application.

## Control of vein patterning by auxin signaling

The residual NPA-insensitive auxin-dependent vein-patterning activity of *pin1,3,6;4;7;8* relies, at least in part, on the signal transduction mediated by the TIR1/AFB auxin receptors and their post-translational regulator AXR1. Loss of *AXR1*; loss of *TIR1* and *AFB2*, the two auxin receptors that most contribute to auxin signaling (*Dharmasiri et al., 2005*); or growth in the presence of the auxin signaling inhibitor PBA (*Matthes and Torres-Ruiz, 2016*) induces entirely new vein-pattern defects in *pin1,3,6;4;7;8* or in its NPA-induced phenocopy. In the more-severely affected leaves of *axr1; pin1,3,6;4;7;8*, *tir1;afb2;pin1,3,6;4;7;8*, NPA-grown *axr1*, NPA-grown *tir1;afb2*, and NPA- and PBA-grown WT, the end-to-end alignment of vascular elements oriented with their axis along the axis of the vein is often replaced by the clustered differentiation of abnormally oriented vascular elements. Not only are these defects never observed in *pin1,3,6;4;7;8* or NPA-grown WT, but they are more severe than the predicted sum of the defects of *pin1,3,6;4;7;8* or NPA-grown WT, on the one hand, and of *axr1*, *tir1;afb2*, or PBA-grown WT, on the other. This synthetic enhancement between the vein pattern defects caused by reduced auxin signaling and those caused by reduced auxin transport suggests non-homologous redundancy of auxin signaling and auxin transport in vein patterning, a conclusion which is consistent with observations in the shoot apical meristem (*Schuetz et al., 2008*). Unlike in the shoot apical meristem, however, in the leaf such redundancy is unequal: whereas auxin transport is required for vein patterning even in the presence of normal auxin signaling, the vein patterning activity of auxin signaling is only exposed in conditions of compromised auxin transport.

How auxin signaling, inherently non-directional (*Leyser, 2018*), could contribute to the polar propagation of the inductive auxin signal in the absence of polar auxin transport is unclear. One possibility is that auxin signaling promotes the passive diffusion of auxin through the tissue by controlling, for example, the proton gradient across the PM (*Fendrych et al., 2016*). However, it is difficult to conceive how auxin diffusion through a specific side of the PM could positively feed back on the ability of auxin to diffuse through that specific side of the PM — a positive feedback that would be required to drain neighboring cells from auxin and thereby form veins, that is channels of preferential auxin movement (*Sachs, 1969*).

One other possibility is that auxin signaling promotes the facilitated diffusion of auxin through the plasmodesmata intercellular channels, a possibility that had previously been suggested (*Mitchison, 1980*) and that has received some experimental support (*Han et al., 2014*). Here, it is conceivable how auxin movement through a specific side of the PM could positively feed back on the ability of the cell to move auxin through that specific side of the PM (e.g., *Cieslak et al., 2015*), but no experimental evidence exists of such feedback or that auxin movement through plasmodesmata controls vein patterning.

Yet another possibility is that auxin signaling activates an unknown mobile signal. Such signal need not be chemical: alternatives, for example a mechanical signal, have been suggested (*Couder et al., 2002*; *Laguna et al., 2008*; *Corson et al., 2009*; *Lee et al., 2014*) and have been implicated in other auxin-driven processes (e.g., *Hamant et al., 2008*; *Heisler et al., 2010*; *Peaucelle et al., 2011*; *Nakayama et al., 2012*; *Braybrook and Peaucelle, 2013*). However, whether a mechanical signal controls vein patterning remains to be tested.

## A tissue-cell-polarizing signal upstream of auxin transport and signaling

The vein pattern defects of leaves in which both auxin transport and signaling are compromised are never observed in leaves in which either process is; yet those defects are not unprecedented: they are observed — though in more extreme form — in leaves of *gn* mutants, suggesting that *GN* controls both auxin transport and signaling during vein patterning.

That *GN* controls PM-PIN-mediated auxin transport during vein patterning is also suggested by the very limited or altogether missing restriction of PIN1 expression domains and coordination of PIN1 polar localization during *gn* leaf development, which is consistent with observations in embryos and roots (*Steinmann et al., 1999*; *Kleine-Vehn et al., 2008*). However, if failure to coordinate the polar localization of PIN1 — and possibly other PM-PIN proteins — were the sole cause of the vein pattern defects of *gn*, these defects would depend on *PM-PIN* function and would therefore be masked by those of *pin1,3;4;7* in the *gn;pin1,3;4;7* mutant. The epistasis of the vein pattern defects of *gn* to those of *pin1,3;4;7* instead suggests that the vein pattern defects of *gn* are independent of *PM-PIN* function; that the vein pattern defects of *gn* are not the sole result of loss or abnormal polarity of PM-PIN-mediated auxin transport; and that *GN* acts upstream of *PM-PIN* genes in vein patterning. Moreover, the epistasis of the vein pattern defects of *gn* to those of *pin1,3,6;4;7;8*, and the inability of NPA, which phenocopies the vein pattern defects of *pin1,3,6;4;7;8*, to induce additional defects in *gn* suggest that the vein pattern defects of *gn* are independent of all the *PIN* genes with vein patterning function; that the vein pattern defects of *gn* are not the sole result of loss or abnormal polarity of PIN-mediated auxin transport; and that *GN* acts upstream of all the *PIN* genes in vein patterning.

Mechanisms by which *GN* controls PM-PIN-mediated auxin transport have been suggested (e.g., *Richter et al., 2010*; *Luschnig and Vert, 2014*; *Naramoto et al., 2014*); it is instead unclear how *GN* could control auxin transport mediated by the ER-localized PIN6 and PIN8. One possibility is that such control depends on *GN* function in ER-Golgi trafficking (*Richter et al., 2007*; *Teh and Moore, 2007*; *Nakano et al., 2009*). Irrespective of the mechanism by which *GN* controls PIN-mediated auxin transport, however, our results suggest that the function of *GN* in coordination of tissue cell polarity and vein patterning entails more than such control, a conclusion which is consistent with functions of *GN* that seem to be unrelated to auxin transport or independent of *PIN* function (*Shevell et al., 2000*; *Fischer et al., 2006*; *Irani et al., 2012*; *Nielsen et al., 2012*; *Moriwaki et al., 2014*).

The auxin-transport-, *PIN*-independent functions of *GN* in coordination of tissue cell polarity and vein patterning are, at least in part, mediated by TIR1/AFB2- and AXR1-mediated auxin signaling. This conclusion is suggested by the ability of simultaneous reduction in auxin transport and signaling to phenocopy defects in coordination of tissue cell polarity, auxin response, and vein patterning of *gn*; it is also supported by the epistasis of the vein pattern defects of *gn* to those of *axr1*, an observation which is consistent with genetic analysis placing *GN* upstream of auxin signaling in the formation of apical-basal polarity in the embryo (*Mayer et al., 1993*).

Though it is unclear how *GN* controls auxin signaling during vein patterning, the most parsimonious account is that GN controls the coordinated localization of proteins produced in response to auxin signaling. Auxin signaling indeed controls the production of proteins that are polarly localized at the plasma membrane of root cells (e.g., *Scacchi et al., 2009*; *Scacchi et al., 2010*; *Yoshida et al., 2019*), and at least some of these proteins act synergistically with PIN-mediated auxin transport in the root (e.g., *Marhava et al., 2018*); however, it remains to be tested whether such proteins have vein patterning activity, whether their localization is controlled by GN, and whether they mediate *GN* function in auxin signaling during vein patterning.

Alternatively, because cell wall composition and properties are abnormal in *gn* (*Shevell et al., 2000*), *GN* may control the production, propagation, or interpretation of a mechanical signal that has been proposed to be upstream of both auxin signaling and transport in the shoot apical meristem (*Heisler et al., 2010*; *Nakayama et al., 2012*); however, whether a mechanical signal controls vein patterning and whether such signal acts downstream of *GN* remains to be tested.

Irrespective of the mechanism of action, our results reveal synergism between auxin transport and signaling, and their unsuspected control by *GN* in the coordination of tissue cell polarity during vein patterning, a control whose logic is unprecedented in multicellular organisms.

## Materials and methods

### Notation

In agreement with *Crittenden et al. (1996)*, linked genes or mutations (<2,500 kb apart, which in Arabidopsis corresponds, on average, to ~10 cM [*Lukowitz et al., 2000*]) are separated by a comma, unlinked ones by a semicolon, and homologous chromosomes by a slash.

### Plants

Origin and nature of lines, and oligonucleotide sequences are in *Supplementary file 1*; genotyping strategies are in *Supplementary file 2Q*. Seeds were sterilized and sown as in *Sawchuk et al. (2008)*. Stratified seeds were germinated and seedlings were grown at 22°C under continuous fluorescent light (~80 μmol m$^{-2}$ s$^{-1}$). Plants were grown at 25°C under fluorescent light (~110 μmol m$^{-2}$ s$^{-1}$) in a 16-h-light/8-h-dark cycle. Plants were transformed and representative lines were selected as in *Sawchuk et al. (2008)*.

### Chemicals

NPA and PBA were dissolved in dimethyl sulfoxide and water, respectively; dissolved chemicals were added (100 μM final NPA concentration, unless otherwise noted) to growth medium just before sowing. IAA was dissolved in melted (55°C) lanolin; the IAA-lanolin paste (1% final IAA concentration) was applied to first leaves 4 days after germination and was reapplied weekly.

### RT-PCR

Total RNA was extracted as in *Chomczynski and Sacchi (1987)* from 4-day-old seedlings grown as in *Odat et al. (2014)*. RT-PCR was performed as in *Odat et al. (2014)* with the following oligonucleotides: 'GN_qFb' and 'GN_qRb', and 'ROC1 F' and 'ROC1 R' (*Beeckman et al., 2002*); 'Aux_F380' and 'Aux_R380', and 'ROC1 F' and 'ROC1 R'; and 'Lax_F513' and 'Lax_R513', and ''ROC1 F' and 'ROC1 R' (*Supplementary file 1*).

### Imaging

Developing leaves were mounted and imaged as in *Sawchuk et al. (2013)*, except that emission was collected from ~2.5 μm-thick optical slices. Light paths are in *Supplementary file 2R*. Mature leaves were fixed in 3 : 1 or 6 : 1 ethanol : acetic acid, rehydrated in 70% ethanol and water, cleared briefly (few seconds to few minutes) — when necessary — in 0.4 M sodium hydroxide, washed in water, mounted in 80% glycerol or in 1 : 2 : 8 or 1 : 3 : 8 water : glycerol : chloral hydrate, and imaged as in *Odat et al. (2014)*. Grayscaled RGB color images were turned into 8-bit images, look-up-tables were applied, and brightness and contrast were adjusted by linear stretching of the histogram in the Fiji distribution (*Schindelin et al., 2012*) of ImageJ (*Schneider et al., 2012*; *Schindelin et al., 2015*; *Rueden et al., 2017*).

## Acknowledgements

We dedicate this article to Ida Ruberti, an inspiring example of scientific depth, integrity, and elegance; Ida passed away on June 8, 2019. We thank the Arabidopsis Biological Resource Center for *emb30-8*, *gn-13*, *gn-18*, *pin1-1*, *eir1-1*, *pin6*, *pin8-1*, *pgp1-100*, *mdr1-101*, *ucu2-4*, *aux1-355*, *lax1-064*, *axr1-3* and *axr1-12*; Hidehiro Fukaki for *fwr* and *gn$^{SALK\_103014}$*; Sandra Richter and Gerd Jürgens for *gn$^{B/E}$* and *gn$^{R5}$*; Satoshi Naramoto and Hiroo Fukuda for *van7/emb30-7*; Eva Benková and Jiří Friml for PIN1::PIN1:GFP; Jian Xu and Ben Scheres for PIN1::PIN1:YFP and PIN2::PIN2:GFP; Jian Xu, Miyo Morita, and Masao Tasaka for PIN3::PIN3:GFP; Ikram Blilou and Ben Scheres for *Atpin1::En134*, *pin3-3*, *pin4-2* and *pin7$^{En}$*; Venkatesan Sundaresan for *toz-1*; Thomas Berleth for *mp$^{G12}$*; Markus Geisler and Jiří Friml for ABCB1::ABCB1:GFP and ABCB19::ABCB19:GFP; Cris Kuhlemeier for *aux1-21;lax1;2–1;3*; Michael Prigge and Mark Estelle for *tir1-1;afb2-3*; and Marcus Heisler and Elliot Meyerowitz for DR5rev::nYFP. This work was supported by Discovery Grants of the Natural Sciences and Engineering Research Council of Canada (NSERC) to ES. CV was supported, in part, by a University of Alberta Doctoral Recruitment Scholarship. MGS was supported, in part, by an NSERC CGS-M Scholarship and an NSERC CGS-D Scholarship.

## Additional information

### Funding

| Funder | Grant reference number | Author |
|---|---|---|
| Natural Sciences and Engineering Research Council of Canada | RGPIN-2016-04736 | Enrico Scarpella |
| University of Alberta | Doctoral Recruitment Scholarship | Carla Verna |
| Natural Sciences and Engineering Research Council of Canada | CGS-M Scholarship | Megan G Sawchuk |
| Natural Sciences and Engineering Research Council of Canada | CGS-D Scholarship | Megan G Sawchuk |

The funders had no role in study design, data collection and interpretation, or the decision to submit the work for publication.

### Author contributions

Carla Verna, Sree Janani Ravichandran, Megan G Sawchuk, Nguyen Manh Linh, Conceptualization, Formal analysis, Validation, Investigation, Visualization, Methodology, Writing—original draft, Writing—review and editing; Enrico Scarpella, Conceptualization, Resources, Formal analysis, Supervision, Funding acquisition, Validation, Investigation, Visualization, Methodology, Writing—original draft, Project administration, Writing—review and editing

### Author ORCIDs

Carla Verna (ID) https://orcid.org/0000-0001-5312-5470
Enrico Scarpella (ID) https://orcid.org/0000-0003-2962-0329

### Decision letter and Author response

Decision letter https://doi.org/10.7554/eLife.51061.sa1
Author response https://doi.org/10.7554/eLife.51061.sa2

## Additional files

### Supplementary files

• Supplementary file 1. Key resources table.

• Supplementary file 2. Supplementary tables.(**A**) Embryo viability of WT, *pin3;4;7*, and *pin2;3;4;7*. (**B**) Embryo viability of *toz*, *mp*, *pin1*, *pin1,3;4;7*, *pin1,3;2;4;7*, and *pin1,3,6;4;7;8*. (**C**) Embryo viability of *pin1*, *pin1,3;4;7*, *pin1,3;2;4;7*, and *pin1,3,6;4;7;8*. (**D**) Embryo viability of WT, *abcb1*, *abcb19*, *abcb1;19*, and *twd1*. (**E**) Embryo viability of *toz*, *mp*, *pin1,3,6*, and *pin1,3,6;abcb1;19*. (**F**) Embryo viability of *pin1,3,6* and *pin1,3,6;abcb1;19*. (**G**) Embryo viability of WT, *aux1*, *lax1*, *aux1;lax1*, and *aux1;lax1;2;3*. (**H**) Embryo viability of *toz*, *mp*, *pin1,3,6*, and *pin1,3,6;aux1;lax1*. (**I**) Embryo viability of *pin1,3,6* and *pin1,3,6;aux1;lax1*. (**J**) Embryo viability of *axr1;axl*, *tir1;afb2*, *gn;pin1,3;4;7*, and *gn;pin1,3,6;4;7;8*. (**K**) Embryo viability of *gn;pin1,3;4;7* and *gn;pin1,3,6;4;7;8*. (**L**) Embryo viability of WT, *axr1*, and *tir1;afb2*. (**M**) Embryo viability of *toz*, *mp*, *pin1,3,6;4;7;8*, *pin1,3,6;4;7;8;axr1*, and *pin1,3,6;4;7;8;tir1;afb2*. (**N**) Embryo viability of *pin1,3,6;4;7;8;axr1* and *pin1,3,6;4;7;8;tir1;afb2*. (**O**) Embryo viability of *toz*, *mp*, *gn* and *gn;axr1*. (**P**) Embryo viability of *gn* and *gn;axr1*. (**Q**) Genotyping strategies. (**R**) Light paths.

• Transparent reporting form

### Data availability

All data generated or analyzed during this study are included in the manuscript and supporting files.

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
