## [Decision Letter]

**Acceptance summary:**

The rearrangement and consolidation of the text has made the manuscript more accessible for the readers. Both the editor and the reviewers applaud the genetic rigor of your study, which provides important novel resources to the field. The conclusions regarding vascular tissue formation represent a profound advance that raises a host of questions to be addressed in the future. We do expect that your work will trigger lively discussions in the field.

**Decision letter after peer review:**

Thank you for submitting your article "Coordination of tissue cell polarity by auxin transport and signaling" for consideration by *eLife*. Your article has been reviewed by two peer reviewers, and the evaluation has been overseen by Christian Hardtke as the Senior and Reviewing Editor.

The reviewers have discussed the reviews with one another and the Reviewing Editor has drafted this decision to help you prepare a revised submission.

As you can see from their comments pasted below, both reviewers are very positive about your manuscript. There are a number of revisions, however, as outlined below, and I would like to ask you to submit a revised manuscript along these lines.

Please pay particular attention to the writing. The way the manuscript is written now it is rather tedious to read and could be much improved by eliminating repetitions, summarising conceptual similar experiments, while streamlining the text at the same time.

Reviewer #1:

Verna et al. report an extremely meticulous analysis of vascular development in a range of mutant combinations that affect the trafficking regulator *GNOM*, its PIN-FORMED trafficking targets, alternative auxin influx and efflux proteins, as well as auxin response mutants.

The *gnom* mutant is an almost iconic mutant with strongly disrupted vascular patterning, hypertrophy of xylem and inability to form continuous vascular strands. The phenotype has always been interpreted as being the consequence of impaired auxin transport. In this manuscript, the authors convincingly demonstrate that altered PIN localization is likely not causal to the *gnom* mutant defect. The analysis is thorough, and involves stacking mutations in nearly all *PIN* genes into the *gnom* mutant. Beyond this, the authors provide evidence that neither ABCB nor AUX//LAX transporters contribute to vascular development in a meaningful way. Finally, the manuscript provides evidence that auxin response contributes to vein formation downstream of *GNOM*.

The work might be considered to be relevant mostly to specialists, but in my perception the findings are important and the genetic analysis is exemplary.

Reviewer #2:

According to a well-accepted model, cell and tissue polarity in developing leaf veins in Arabidopsis, is established by polar transport of the hormone auxin, the consequence of asymmetric localization of PIN transporters. In the vein and other organs/tissues PIN localization requires the *GNOM* GEF. In this study the authors systematically test this model through a very careful genetic analysis using mutants affected in various auxin transport and signaling genes. As previously shown, they show that gn mutants have a very strong polarity defect in leaf veins. Surprisingly, they find that higher order mutants in *PIN* genes, as well as other genes that function in auxin transport have a vein phenotype that is significantly less severe than gn implying that the PINs are not essential to establish polarity in the veins. Further the show that mutations that affect auxin signaling or conditions that inhibit auxin signaling, result in a GN-independent defect in vein polarity that is independent of the PINs. These are intriguing results that should force a re-examination of the prevailing model of vein development as well as other instances where PIN activity is thought to drive polarity establishment. I am very enthusiastic about publication after the following issues have been addressed.

1) The paper lacks flow and is difficult to read. Part of this is because the experiments involve many somewhat complex genotypes. However, the presence of many repeated phrases and general repetitive style sometimes make for rough going. In other cases, there is a lack of clarity. These are important results and I think the authors should work to make them more accessible.

2) I think it would be useful to have a figure that illustrates the dynamic movement of the PINs during vein development.

3) The authors’ interpretation of their results depends on the fact that many of the alleles employed are nulls (particularly the *pin* alleles).

4) In some cases, it would be useful if conclusions were stated more clearly. For example, instead of stating that such and such as combination of alleles do not phenocopy *gnom*, they could say that the phenotype of this allele combination is much less severe than gn, implying that GN is doing something besides mediating PIN localization.

5) Is it correct to say that GN is "controlling" PIN localization, rather than being required for this process?

6) In the subsection “Contribution of the *GNOM* Gene to Coordination of Tissue Cell Polarity During Arabidopsis Vein Formation”, the authors introduce several different gn alleles. What are we to take from this?

---

## [Author Response]

As you can see from their comments pasted below, both reviewers are very positive about your manuscript. There are a number of revisions, however, as outlined below, and I would like to ask you to submit a revised manuscript along these lines.Please pay particular attention to the writing. The way the manuscript is written now it is rather tedious to read and could be much improved by eliminating repetitions, summarising conceptual similar experiments, while streamlining the text at the same time.

We thank the Editors for their constructive feedback. As detailed below in our response to the reviewers’ comments, we have restructured the Results section under new headings and sub-headings to bring out the connection between its different parts. Further, we have shortened and focused all the sections in the manuscript – in particular, we have shortened the Results section by ~40%; some parts were moved, some were ruthlessly deleted. Finally, we have consolidated the original 13 core figures into 10 revised ones.

Reviewer #2:[…] I am very enthusiastic about publication after the following issues have been addressed.1) The paper lacks flow and is difficult to read. Part of this is because the experiments involve many somewhat complex genotypes. However, the presence of many repeated phrases and general repetitive style sometimes make for rough going. In other cases, there is a lack of clarity. These are important results and I think the authors should work to make them more accessible.

We thank reviewer 2 for their stimulating feedback, based on which we have made the following changes.

- We have presented the hypothesis and its predictions only once, at the beginning of the Results section, under the heading “Testable Predictions of the Current Hypothesis of Coordination of Tissue Cell Polarity and Vein Formation by Auxin”; this solution eliminates unnecessary repetitions.

- We have revised headings to describe better what the reader will find in each sub-section of the Results; this solution provides structure and moves the text at a steady pace in the direction the text is supposed to go.

- We have focused on results as opposed to data; we have moved information that is peripheral to the main message of the manuscript; and we have deleted whatever was not essential to that message; this solution has enabled us to sharpen the focus of the manuscript.

The combination of these changes has enabled us to shorten the Results section by ~40%; it has also allowed shortening, though to a lesser extent, the Introduction and Discussion sections.

2) I think it would be useful to have a figure that illustrates the dynamic movement of the PINs during vein development.

We agree with reviewer 2 that understanding the expression and localization during vein development of *PIN2, PIN3, PIN4*, and *PIN7* (we assume that is what reviewer 2 had in mind – *PIN1* expression and localization have in fact already been the focus of at least four papers: Scarpella et al., 2006; Wenzel et al., 2007; Bayer et al., 2009; Marcos and Berleth, 2014) would be interesting; however, as clarified below, we believe such understanding does not belong in this manuscript.

The goal of our study was to test the prevailing hypothesis of coordination of tissue cell polarity and vein formation by auxin. Simultaneous mutation in all the *PIN* genes with vein patterning function – including *PIN3, PIN4*, and *PIN7* – fails to lead to defects that fall within the vascular phenotype spectrum of *gn*. Furthermore, the vein pattern phenotype induced by simultaneous mutation in all the *PIN* genes with vein patterning function – including *PIN3, PIN4*, and *PIN7* – fails to be epistatic to the *gn* vascular phenotype. Therefore, abnormal expression and localization of *PIN1* – and of all the other *PIN* proteins with vein patterning function, including *PIN3, PIN4*, and *PIN7* – cannot be the cause of the *gn* vascular phenotype. As such, the expression and localization of *PIN3, PIN4*, and *PIN7* during vein development – as interesting as they may be in themselves – are irrelevant to the goal of testing the prevailing hypothesis of coordination of tissue cell polarity and vein formation by auxin.

As for *PIN2*, the normal vein pattern of *pin2* (Sawchuk et al., 2013), and the inability of *pin2* to enhance the vein pattern phenotype of *pin1* (Sawchuk et al., 2013) and *pin1,3;4;7* (this manuscript) suggest that *PIN2* expression in the leaf epidermis is inconsequential for vein patterning and thus beyond the scope of our study.

Incidentally, a description of the different aspects of the expression and localization of *PIN1* during vein development has required at least four manuscripts (Scarpella et al., 2006; Wenzel et al., 2007; Bayer et al., 2009; Marcos and Berleth, 2014); an acceptable study of the expression and localization of *PIN2, PIN3, PIN4*, and *PIN7* during vein development cannot require fewer than four figures. As such, we would like to suggest that the addition of four figures to a manuscript that already has – after reduction – 10 information-rich figures, 1 summary figure, 23 figure supplements, and 19 supplementary files (now consolidated into two supplementary files) would only be justified if the new evidence were essential to the main conclusion of the manuscript, which – as detailed above – in the case of our manuscript we believe it is not.

3) The authors’ interpretation of their results depends on the fact that many of the alleles employed are nulls (particularly the pin alleles).

We thank reviewer 2 for the opportunity to clarify this important concept. The alleles used in our study have been well-characterized elsewhere (references in Supplementary file 1 – Key Resources Table) or in this manuscript, including the newly added Figure 5—figure supplement 3, and are either null or the strongest alleles available (except, of course, for the weak and intermediate alleles of *gn* and for *axr1-3*, which are partial loss-of-function alleles and were used precisely for that reason). Most important, however, our conclusions do not depend on complete loss of gene function. For example, were the *gn* vascular phenotype caused by loss of *PIN*-mediated polar auxin transport, simultaneous mutation of *PIN* genes with vein patterning function would lead to defects that fall within the vascular phenotype spectrum of *gn*. This would be the case irrespective of whether those *pin* mutations led to complete loss of function or to partial loss of function: in the former case, the vascular phenotype of that multiple *pin* mutant would resemble that of strong *gn* alleles; in the latter, it would resemble that of weaker *gn* alleles. Neither prediction is supported by our results; nevertheless, we realize our original text was misleading. We have now revised it to explain more clearly the testable predictions of the hypothesis and the conclusions derived from the results of the tests of those predictions.

4) In some cases, it would be useful if conclusions were stated more clearly. For example, instead of stating that such and such as combination of alleles do not phenocopy gnom, they could say that the phenotype of this allele combination is much less severe than gn, implying that GN is doing something besides mediating PIN localization.

We thank reviewer 2 for their suggestion, which we took into account as we revised the Results section according to the new structure inspired by the comments of reviewer 2 (see our response to comment no. 1 of reviewer 2).

5) Is it correct to say that GN is "controlling" PIN localization, rather than being required for this process?

We believe both of them are correct. Examples abound in literature of the use of the term “control” in the way we have used it; a recent example is the report that retromer in *Drosophila* controls the localization of core planar polarity proteins (Strutt et al. 2019. Curr Biol. 29: 484–491). In that study, just as in ours, such control is inferred from abnormal protein localization in loss-of-function mutants.

6) In the subsection “Contribution of the GNOM Gene to Coordination of Tissue Cell Polarity During Arabidopsis Vein Formation”, the authors introduce several different gn alleles. What are we to take from this?

We realize we had not been clear about this, and we appreciate the opportunity reviewer 2 has given us to clarify this important point. As it should now be clear from the revised Results section – and as also mentioned in our response to reviewer 2’s comment no. 3 – the prevailing hypothesis of coordination of tissue cell polarity and vein formation by auxin predicts that, were the vein pattern defects of *gn* the result of loss of polar auxin transport, auxin transport inhibition would lead to defects that fall within the vascular phenotype spectrum of *gn*. Therefore, to test this prediction it is essential to understand what the vascular phenotype spectrum of *gn* is; and to understand what the vascular phenotype spectrum of *gn* is, it is essential to determine the vascular phenotype of an allelic series in the *GN* gene, which is what we had presented in Figure 2 and had described in the text reviewer 2 refers to.